# Slx5/Slx8-dependent ubiquitin hotspots on chromatin contribute to stress tolerance

Markus Höpfler[1] (iD), Maximilian J Kern[1], Tobias Straub[2] (iD), Roman Prytuliak[3], Bianca H Habermann[3,4] (iD), Boris Pfander[5,*] (iD) & Stefan Jentsch[1,†]

## Abstract

Chromatin is a highly regulated environment, and protein association with chromatin is often controlled by post-translational modifications and the corresponding enzymatic machinery. Specifically, SUMO-targeted ubiquitin ligases (STUbLs) have emerged as key players in nuclear quality control, genome maintenance, and transcription. However, how STUbLs select specific substrates among myriads of SUMOylated proteins on chromatin remains unclear. Here, we reveal a remarkable co-localization of the budding yeast STUbL Slx5/Slx8 and ubiquitin at seven genomic loci that we term "ubiquitin hotspots". Ubiquitylation at these sites depends on Slx5/Slx8 and protein turnover on the Cdc48 segregase. We identify the transcription factor-like Ymr111c/Euc1 to associate with these sites and to be a critical determinant of ubiquitylation. Euc1 specifically targets Slx5/Slx8 to ubiquitin hotspots via bipartite binding of Slx5 that involves the Slx5 SUMO-interacting motifs and an additional, novel substrate recognition domain. Interestingly, the Euc1-ubiquitin hotspot pathway acts redundantly with chromatin modifiers of the H2A.Z and Rpd3L pathways in specific stress responses. Thus, our data suggest that STUbL-dependent ubiquitin hotspots shape chromatin during stress adaptation.

**Keywords** Cdc48/p97; chromatin remodeling; STUbL; SUMO; ubiquitin
**Subject Categories** Chromatin, Epigenetics, Genomics & Functional Genomics; Post-translational Modifications, Proteolysis & Proteomics
The EMBO Journal (2019) 38: e100368

## Introduction

SUMO-targeted ubiquitin ligases (STUbLs) modify SUMOylated proteins with ubiquitin and thereby transfer substrates from the SUMO (small ubiquitin-like modifier) to the ubiquitin pathway (Sriramachandran & Dohmen, 2014). To achieve this, STUbLs combine binding to SUMOylated proteins via SUMO-interacting motifs (SIMs) with ubiquitin ligase activity (Prudden *et al*, 2007; Sun *et al*, 2007; Uzunova *et al*, 2007; Xie *et al*, 2007). Apart from this defining feature, the STUbL enzyme family is highly heterogeneous, as is the regulation of each member, even though functional aspects appear to be conserved (Sriramachandran & Dohmen, 2014). Of note, thousands of proteins are SUMOylated in cells (Hendriks & Vertegaal, 2016), but only a handful of them were shown to be targeted by STUbLs. This raises the question of which features make a protein a STUbL substrate.

A hallmark of several STUbL substrates is modification with SUMO chains (polySUMOylation) (Uzunova *et al*, 2007; Tatham *et al*, 2008), but it has also been suggested for the human STUbLs RNF4 and Arkadia/RNF111, as well as for *Drosophila* Degringolade/ Dgrn that they might use additional SUMO-independent interactions for substrate recognition (Abed *et al*, 2011; Groocock *et al*, 2014; Kuo *et al*, 2014; Sun *et al*, 2014; Thomas *et al*, 2016). However, in case of the prototypical STUbL, budding yeast Slx5/Slx8, no substrate recognition elements have been characterized other than its SUMO-interacting motifs.

STUbLs orchestrate many nuclear functions such as, but not limited to, DNA repair, quality control, and transcriptional regulation (Sriramachandran & Dohmen, 2014). Accordingly, most STUbL substrates are nuclear proteins. Human RNF4, for example, targets the PML (promyelocytic leukemia) protein, which is polySUMOylated in nuclear PML bodies upon arsenic exposure (Tatham *et al*, 2008). Other RNF4 substrates include transcription factors and proteins involved in different DNA repair pathways (for reviews, see Sriramachandran & Dohmen, 2014; Nie & Boddy, 2016). Budding yeast Slx5/Slx8 was initially identified for its role in genome stability as well, which manifests in a synthetic lethal phenotype with the DNA helicase Sgs1 (Mullen *et al*, 2001). Slx5/Slx8 is involved in the repositioning of DNA lesions to nuclear pore complexes (Nagai *et al*, 2008; Su *et al*, 2015; Churikov *et al*, 2016; Horigome *et al*, 2016). In line with an additional major function in chromatin maintenance, several DNA-associated proteins have been described as Slx5/Slx8 substrates (Ohkuni *et al*, 2016; Schweiggert *et al*, 2016; Thu *et al*, 2016; Liang

1 Max Planck Institute of Biochemistry, Molecular Cell Biology, Martinsried, Germany
2 Biomedizinisches Centrum, Core Facility Bioinformatics, Ludwig-Maximilians-Universität München, Martinsried, Germany
3 Max Planck Institute of Biochemistry, Computational Biology Group, Martinsried, Germany
4 Aix-Marseille Univ, CNRS, IBDM UMR 7288, Marseille Cedex 9, France
5 Max Planck Institute of Biochemistry, DNA Replication and Genome Integrity, Martinsried, Germany
*Corresponding author. Tel: +49 89 85783050; E-mail: bpfander@biochem.mpg.de
†Deceased during the course of this study

*et al*, 2017), including transcription factors (TFs) such as Mot1 (mutant variant) and Matα2 (Wang & Prelich, 2009; Xie *et al*, 2010). Interestingly, in the latter case, Matα2 SUMOylation is dispensable for Slx5/Slx8 targeting, but the SIMs of Slx5 and Matα2 DNA binding are required (Xie *et al*, 2010; Hickey *et al*, 2018). Matα2 ubiquitylation subsequently facilitates the recruitment of the Cdc48 complex (p97/VCP in mammalian cells) (Wilcox & Laney, 2009), a segregase that can extract ubiquitylated proteins from their local environment, such as chromatin (Rape *et al*, 2001; Ramadan *et al*, 2007; Maric *et al*, 2014; Franz *et al*, 2016).

It emerges that both sequence-specific DNA-binding proteins and other chromatin-associated proteins are STUbL substrates. However, it is still unknown whether STUbLs fulfill a general role in regulating protein turnover at chromatin and to what extent other components of the ubiquitin–proteasome system (UPS), such as Cdc48, are involved. To address these questions, we obtained genome-wide binding profiles of Slx8 and ubiquitylated proteins. Notably, Slx8 localized to surprisingly few genomic sites, and the ubiquitin signal at seven of these sites was Slx5/Slx8-dependent and strongly enriched in *cdc48* mutant strains. These data indicate that these "ubiquitin hotspots" are sites of STUbL- and Cdc48-dependent protein turnover on chromatin. Ubiquitin hotspots are bound by the poorly characterized transcription factor-like protein Ymr111c/Euc1, which is modified with SUMO and is a STUbL substrate. Notably, however, deletion of *EUC1* does apparently not lead to an abrogation of transcription in the vicinity of ubiquitin hotspots, but rather results in strong genetic interactions with H2A.Z and Rpd3 pathways, which regulate expression of many genes. Euc1 and ubiquitin hotspots are part of an Rpd3S-dependent pathway that is required to cope with cellular stress induced by suboptimal temperature. Moreover, the analysis of the Slx5/Slx8-recruitment mechanism led to the identification of a SUMO-independent substrate-binding domain within Slx5, suggesting a new mode of substrate recognition by Slx5/Slx8.

# Results

### Slx5/Slx8 and Cdc48 control seven ubiquitin hotspots across the yeast genome

To investigate Slx5/Slx8-catalyzed ubiquitylation of chromatin-associated proteins, we developed chromatin immunoprecipitation (ChIP) protocols for ubiquitylated proteins (Appendix Fig S1A) and Slx5/Slx8 in *Saccharomyces cerevisiae*. We used the FK2 ubiquitin antibody with broad specificity toward mono-ubiquitylated proteins and K29-, K48-, and K63-linked chain types combined with genome-wide tiling arrays (ChIP-chip, Fig 1A). We detected ubiquitin signals at open reading frames (ORFs). These signals appear to represent histone H2B mono-ubiquitylation that has been described to be particularly abundant on highly transcribed genes (Braun & Madhani, 2012), as they were either lost in cells that harbor a mutation of the main H2B ubiquitylation site (*h2b-K123R*) or in cells that lack Rad6, the E2 enzyme for H2B ubiquitylation (Jentsch *et al*, 1987; Robzyk *et al*, 2000) (Fig 1A and Appendix Fig S1B, Dataset EV1). However, some ubiquitin signals were H2B ubiquitylation-independent and notably enhanced in *cdc48* mutants (*cdc48-6*, Fig 1A, Dataset EV1).

In similar experiments, Slx8 bound specifically to only few sites in the genome (*Slx8-9myc*, Fig 1B, Dataset EV2). Comparison of our genome-wide profiles of regions enriched for both ubiquitin- and Slx8-binding revealed a striking correlation for seven sites, which we term "ubiquitin hotspots" (ub-hotspots, ub-HS in figures, Figs 1B and C, and EV1A and B, Datasets EV1–EV3). Besides these seven "ubiquitin hotspots", we detected only two sites of major ubiquitin accumulation in *cdc48* mutants without Slx8 enrichment (ub-only-sites), and two distinct sites of major Slx8 enrichment without ubiquitin accumulation (Datasets EV1 and EV2, see also Appendix Fig S2E).

Next, we determined the accumulation of specific ubiquitin chain types at ub-hotspots. Using a ubiquitin K48 chain-specific antibody (clone Apu2, "ub-K48"), we detected the same ub-hotspots as with the FK2 antibody (Fig EV1A, Dataset EV3). Moreover, strains harboring different mutant alleles of the *CDC48* gene showed an increase of ubiquitin-ChIP signals from around 5–10-fold (WT) to 15–50-fold (*cdc48-6*, *cdc48-3*) enrichment over background, with comparable results for both antibodies (Fig EV1C and D). Since the ub-K48 antibody was more specific for the Slx8-bound and Cdc48-controlled ub-hotspots, we used this antibody for the rest of our study.

Consistent with recruitment of the Slx5/Slx8 heterodimer to chromatin, we found that both subunits were enriched to similar levels at selected ub-hotspots, and observed a moderate increase in Slx5/Slx8 binding in *cdc48-3* mutant cells (Fig 1D). While the ubiquitin signals did not correlate with Slx5/Slx8 enrichment levels at the tested sites (compare Fig 1D and E), they dropped to background levels in *slx5Δ* and *slx8Δ* cells (Fig 1F), indicating that Slx5/Slx8 is the relevant ubiquitin E3 ligase at these sites.

Cdc48 targeting is usually facilitated by cofactors that mediate substrate specificity (Buchberger *et al*, 2015). Consistent with previous results for other chromatin-bound substrates (Verma *et al*, 2011), we found that specifically the Cdc48$^{Ufd1-Npl4}$ complex removes ubiquitylated proteins from ub-hotspots (Figs 1G and EV1E), assisted by additional Ubx4 and Ubx5 cofactors (Fig EV1E and Appendix Fig S1C, Dataset EV3). In contrast, impairment of Cdc48 substrate delivery to the proteasome (*rad23Δ dsk2Δ*; Richly *et al*, 2005) or proteasome assembly (*ump1Δ*; Ramos *et al*, 1998) did not cause accumulation of ubiquitin conjugates at ub-hotspots (Appendix Fig S1D).

Taken together, our analysis of genome-wide ubiquitin and Slx8 ChIP data reveals seven ubiquitin hotspots that share similar features: (i) strong accumulation of ubiquitin in *cdc48* and associated cofactor mutants, (ii) recruitment of Slx5 and Slx8, and (iii) dependence on the functional Slx5/Slx8 dimer for ubiquitylation.

### A sequence motif within ub-hotspots is bound by Ymr111c/Euc1

Interestingly, all ub-hotspots lie within intergenic regions, do not seem to be associated with any annotated features within the yeast genome, and appear to be distributed among the sixteen yeast chromosomes (Fig 1C). We could also not identify any shared pathway or function of the adjacent genes (Dataset EV4).

To investigate whether any sequence features define ub-hotspots, we cloned a 1,038-bp region of ub-HS4 on chromosome XIII (*ChrXIII*) and inserted it into the *LEU2* locus on *ChrIII* (ectopic ub-HS4, Fig 2A). This fragment was sufficient to drive formation of an

ectopic ub-hotspot at the new position (Appendix Fig S2A), suggesting a role for specific DNA sequences. Indeed, we were able to map a minimal 39-bp fragment required for ub-hotspot formation (*ub-HS4 F7*, Appendix Fig S2A, Fig 2A–C).

We also used the MEME suite to predict sequence motifs within the ub-hotspots (Bailey *et al*, 2009). Consistent with our experimental mapping, a sequence motif of 36 bp was identified, which largely overlapped with the experimentally mapped 39-bp fragment (Fig 2C). This "ub-HS-motif" was found in all the ub-hotspots in at least one copy (Appendix Fig S2B), suggesting that a sequence-specific DNA-binding protein might localize to ub-hotspots. Supporting this notion, a single point mutation within one of the central,

conserved TTGTT repeats led to a complete loss of ubiquitin from the ectopic ub-hotspot (*ub-HS4 F7-mut*, Fig EV1F).

To identify proteins binding to the ub-HS-motif, we applied an unbiased yeast one-hybrid (Y1H) screening strategy (Fig 2D). In two independent screens in either a WT or a *ubx5Δ* strain, which shows enriched ubiquitin at ub-hotspots (Fig EV1E), we identified several clones of four different genes: *SMT3* (encoding for SUMO), *SLX5*, and the uncharacterized *YMR111C/EUC1* and *YFR006W* (Figs 2D and EV1G). Identification of SUMO suggests a SUMOylation event upstream of Slx5/Slx8 recruitment, while isolation of *SLX5* clones confirms our ChIP data (Fig 1D). We confirmed the recruitment of Ymr111c/Euc1, SUMO, and Yfr006w with Gal4

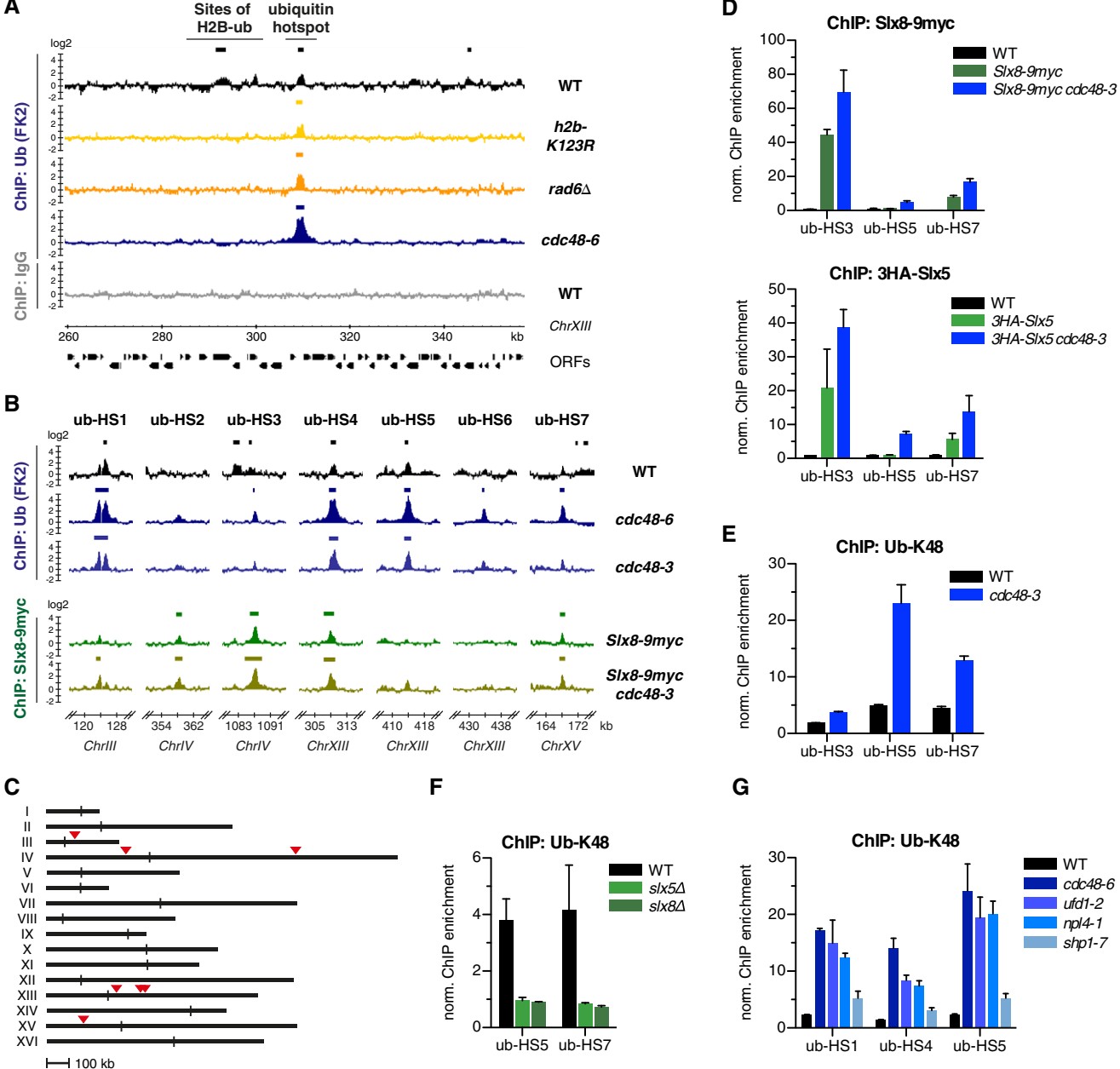

**Figure 1.**

◀

**Figure 1.  The SUMO-targeted ubiquitin ligase Slx5/Slx8 is required for the formation of seven ubiquitin hotspots across the yeast genome.**

A  Genome-wide ubiquitin binding profiles identify numerous regions of histone H2B ubiquitylation in WT cells and distinct sites of non-H2B ubiquitylation ("ubiquitin hotspot", ub-hotspot, ub-HS) which persist in *h2b-K123R* and *rad6Δ* strains, and increase in *cdc48* mutants (*cdc48-6*). A 90 kb stretch of chromosome XIII (*ChrXIII*) is depicted. Chromatin immunoprecipitation was performed using the FK2 ubiquitin antibody (see also Appendix Fig S1A), and enriched DNA was analyzed on NimbleGen arrays (Chip-chip). DNA from non-specific IgG ChIP experiments served as background control. Significantly enriched regions are marked by bars above the respective ChIP-chip tracks and are summarized in Dataset EV1. Data represent means from two independent replicates, except for ubiquitin (FK2) in *rad6Δ* and IgG-ChIP in WT (*n* = 1). All experiments, including those using *cdc48-6* and other temperature-sensitive (ts) alleles, were performed at 30°C (semi-permissive temperature for ts-alleles) unless stated otherwise.

B  Seven ub-HSs show strong correlation between ubiquitin binding and Slx8 enrichment (Slx8-9myc ChIP). 16-kb windows of the indicated regions centered around the ub-HSs are depicted for ubiquitin and Slx8 binding profiles. Ubiquitin (FK2) data for WT and *cdc48-6* are from the same experiment as depicted in (A). Data represent means from two independent replicates.

C  Schematic representation of the 16 yeast chromosomes (I–XVI) with positions of the ub-HSs marked by red triangles. Vertical bars indicate positions of centromeres.

D  Slx8 and Slx5 are recruited to ub-HSs. DNA from Slx8-9myc (top) or 3HA-Slx5 (bottom) ChIP experiments of the indicated strains was analyzed by quantitative real-time PCR (ChIP-qPCR) at selected ub-HSs. Data represent mean ± standard deviation (SD, *n* = 2). Consistent with a previous study, we did not observe any Slx8 enrichment at centromeres (van de Pasch *et al*, 2013); however, we note that we could not confirm the reported centromere binding of Slx5 (see also Appendix Fig S2E).

E  K48-linked ubiquitin chains accumulate in a *cdc48* mutant (*cdc48-3*). Ub-K48 ChIP followed by qPCR for the same loci as in (D) is depicted. See also Fig EV1C and D. Data represent means ± SD (*n* = 2).

F  Slx5 and Slx8 are required for ub-HS formation. Ub-K48 ChIP-qPCR in strains lacking one subunit of the Slx5/Slx8 complex (*slx5Δ*, *slx8Δ*). Data represent means ± SD (*n* = 3).

G  Ufd1 and Npl4 act in concert with Cdc48 to remove ubiquitylated proteins from ub-HS sites. Ub-K48 ChIP for the indicated strains grown at the semi-permissive temperature of 30°C. Data represent means ± SD (*n* = 2).

Data information: All ChIP-qPCR data represent means ± SD from 2 to 5 independent experiments as indicated, with quantification in triplicates. Data were normalized to an unrelated control region on *ChrII* (see Materials and Methods).

activation domain (AD) fusion fragments (Fig 2E and Appendix Fig S2C). Importantly, deletion of *YMR111C/EUC1* led to a complete loss of ubiquitin-ChIP signals at ub-hotspots, while deletion of *YFR006W* had no effect (Fig 2F and G, Dataset EV3). Therefore, we here name *YMR111C* as "*EUC1*" (*Enriches Ubiquitin on Chromatin 1*).

Consistent with a key role of Euc1 in the formation of ub-hotspots, we found that activation of the *HIS3* reporter by AD-SUMO was Euc1-dependent (Appendix Fig S2C), suggesting that Euc1 binding occurs before SUMO binding or a SUMOylation event. In line with this, Euc1 could not bind the mutated ub-HS4 sequence (Appendix Fig S2D).

We raised an antibody against Euc1 to test its association with the endogenous ub-hotspot sites in ChIP-chip experiments (Fig 2H, Dataset EV5). As expected from the Y1H assays, Euc1 strongly accumulated at ub-hotspots in WT but not in *euc1Δ* cells (Fig 2H), nor at the mutated ectopic ub-HS4 sequence (Fig EV1H). Notably, ub-hotspots are the major sites of Euc1 binding in the entire genome, with only three additional sites of Euc1 accumulation (two of which also enrich Slx8 and contain the ub-HS-motif, Appendix Fig S2E). These data indicate that Euc1 specifically localizes to ub-HS-motif sites and is required for the formation of the ub-hotspots.

### The transcription factor-like Euc1 shows transactivation in reporter gene assays

Euc1 harbors a predicted coiled-coil (CC) domain in its N-terminal part and a GCR1 domain at its C-terminus (Fig 3A). GCR1 domains have been shown to confer sequence-specific DNA binding in Gcr1 and related transcription factors (TFs) (Huie *et al*, 1992; Hohmann, 2002). A distantly related GCR1 domain protein that also binds DNA is Cbf2, which is part of the CBF3 complex and establishes kinetochore attachment with centromeres (Espelin *et al*, 2003). Protein structure prediction suggested a myb-like DNA-binding fold within the GCR1 domain (Appendix Fig S3A) (Biedenkapp *et al*, 1988; Kelley *et al*, 2015) and our mapping results confirm that the

complete GCR1 domain and C-terminus are essential for Euc1 association with the ub-HS-motif (Appendix Fig S3B). Introduction of two point mutations to the predicted DNA-binding loop (W333A, R334A, *euc1-DBD\**) resulted in complete loss of association with ub-hotspots in Y1H assays (Appendix Fig S3C) and Euc1 ChIP experiments (Appendix Fig S3D and E), suggesting that Euc1 directly binds the ub-HS-motif. Concomitantly, ubiquitin enrichment at ub-hotspots was also lost in *euc1-DBD\** cells (Appendix Fig S3E).

Phylogenetic analysis revealed *EUC1*-like genes in several other yeast species, with most pronounced sequence homology in CC- and GCR1 domains (Fig 3A, Appendix Fig S4A). Conversely, ub-hotspot sequences could be identified in those *Saccharomyces* species, where corresponding intergenic regions could be aligned to *S. cerevisiae* (Fig 3B, Appendix Fig S4B). We note that residues most conserved in the different hotspot motifs also appeared most highly conserved in related yeasts, hinting at a similar function of Euc1 proteins at these sites.

To test whether Euc1 itself could also function as a transcriptional activator, we deleted endogenous *EUC1* in the *ub-HS4-HIS3 ubx5Δ* Y1H reporter strain, which reduced background activation of the *HIS3* reporter (Appendix Fig S4C). Conversely, plasmid-borne expression of Euc1 using its endogenous promoter was sufficient to drive transcription of the reporter (Fig 3C and D). To map the transactivation domain of Euc1, we introduced truncations in the N-terminus that contains an acidic patch (aa 19–28, Appendix Fig S4D) (Sigler, 1988). Consistent with a role in transactivation, truncation of either the first 15 or 30 amino acids led to a strong decrease in *HIS3* expression (Fig 3C). Moreover, the first 30 amino acids of Euc1, when fused to the Gal4 DNA-binding domain (BD), were sufficient to drive expression of *HIS3* under the control of a *GAL1* promoter (Appendix Fig S4D). While these data suggest that Euc1 could act as a transcription factor at ub-hotspots, they are also consistent with a model whereby Euc1 mediates ub-hotspot formation to establish a specific chromatin domain or structure, like Cbf2 does at centromeres (see below).

**Ub-hotspot formation requires Euc1 SUMOylation**

Our initial data suggested that SUMOylation (Appendix Fig S2C) and ubiquitylation (Fig 2F and G) might be crucial regulators of the ub-hotspots. We considered Slx5/Slx8 as the prime candidate for the responsible ubiquitin ligase (Fig 1B, D and F) and we set out to investigate how this highly specific recruitment would be established and regulated. Several large-scale proteomic studies reported Ymr111c/Euc1 as a putative SUMOylation substrate (Zhou *et al*, 2004; Denison *et al*, 2005; Hannich *et al*, 2005) and we found Euc1 to interact with conjugatable SUMO (SUMO-GG) as well as the SUMOylation enzymes Ubc9 (E2), Siz1, and Siz2 (E3s) in a yeast

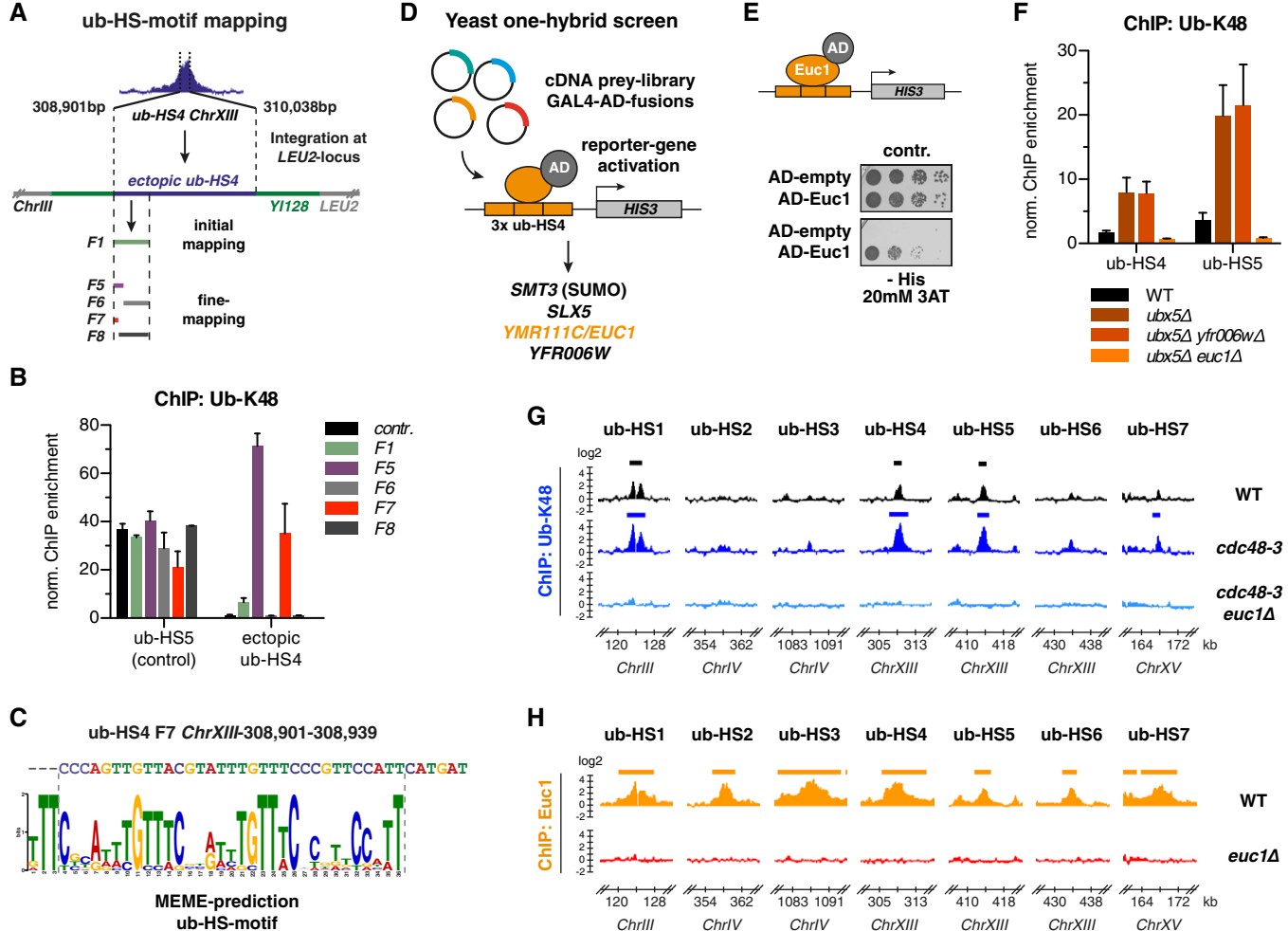

**Figure 2. A sequence motif within ub-hotspots is bound by Ymr111c/Euc1.**

A    Schematic of the ub-HS-motif mapping strategy. A 1,038-bp stretch of ub-HS4 (blue line) was cloned and integrated at the *LEU2* locus (gray) using the integrative YIplac128 vector (green). Initial mapping led to the identification of fragment F1 (Appendix Fig S2A), which was further truncated for fine-mapping ((B), F5–F8). qPCR primers were designed to bind within the YI128 backbone.

B    A 39-bp fragment of ub-HS4 is sufficient to drive ectopic ub-HS formation. Fragments of ub-HS4 were integrated ectopically, and ub-K48 ChIP-qPCR was performed for the endogenous ub-HS5 and the ectopic ub-HS4 fragments as depicted in (A). contr.: control, empty YIplac128 vector was integrated at *LEU2*. Experiments were performed in *cdc48-6* background. Data represent means ± SD (*n* = 2).

C    Experimental mapping and bioinformatic prediction identify a similar ub-HS-motif. Comparison between the experimentally mapped ub-HS4 F7 (B) and the consensus motif of all ub-HSs identified by the MEME software.

D    A yeast one-hybrid screening strategy to identify proteins binding to the ub-HS-motif. Three copies of the ub-HS4-motif (F7) were cloned upstream of a minimal promoter followed by a *HIS3* reporter gene and integrated at the *URA3* locus. A yeast cDNA library with N-terminally fused Gal4 activation domain (AD) was used to screen for survival on media lacking histidine and multiple plasmids coding for 4 different genes were recovered (bottom). See also Fig EV1G.

E    Euc1 binds to the ub-HS-motif in a Y1H assay. Gal4-AD- or Gal4-AD-Euc1-encoding plasmids were transformed into a Y1H reporter strain as described in (D). Serial dilutions were spotted on control plates and plates lacking histidine with 20 mM 3-amino-triazole (3AT). Cells were grown at 30°C for 3 days.

F    Euc1 is required for the formation of ub-HSs. ChIP with ub-K48 specific antibodies was performed for the indicated strains, and enriched DNA was analyzed by qPCR. Data represent means ± SD (*n* = 4).

G, H   Euc1 binds to endogenous ub-HSs. Genome-wide binding profiles of K48-linked ubiquitin chains (G) or Euc1 (H) were obtained in ChIP-chip experiments as described in Fig 1A. Euc1-ChIP experiments were performed with a polyclonal antibody raised against Euc1 aa 292–462. Data represent means from two independent replicates.

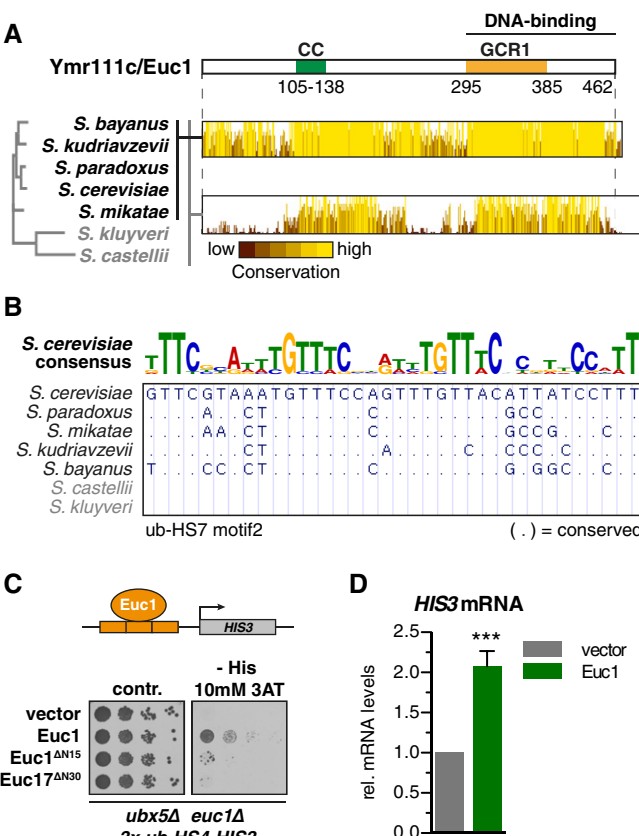

**Figure 3. The transcription factor-like Euc1 shows transactivation in reporter gene assays.**

A Euc1 domain structure is reminiscent of transcription factors. Top: Schematic representation of Euc1, with a predicted coiled-coil domain (CC) and GCR1 domain indicated. Predicted DNA binding of the C-terminal part was confirmed in Y1H experiments (Appendix Fig S3A–C). Bottom: Euc1-like protein sequences from closely related *Saccharomyces* species were aligned, and a phylogenetic tree was generated using Clustal Omega. Jalview was used to graphically display the degree of sequence conservation (see also Appendix Fig S4A).

B The ub-HS-motif is conserved in closely related yeast species. The 7 yeast Multiz Alignment & Conservation tool of the UCSC Genome Browser was used to retrieve alignments of sequences corresponding to ub-HS-motifs from other *Saccharomyces* species. Dot (.) indicates a conserved base.

C Euc1 can induce transactivation via its N-terminal 30 amino acids. Euc1 constructs under the endogenous *EUC1* promoter were transformed in a reporter strain as described for Fig 2D, and serial dilutions were spotted on control or selective media to test *HIS3* activation. Cells were grown at 30°C for 3 days.

D Quantification of *HIS3* mRNA levels from strains used in (C). Cells were grown in liquid media with selection for the transformed plasmids (SC-Leu), harvested in logarithmic growth phase, and total mRNA was prepared. After reverse transcription, *HIS3* mRNA levels were quantified using qPCR (RT–qPCR), normalized first to *ACT1* mRNA and then to the empty vector control strain. Data represent means ± SD ($n = 4$). $P = 2.43 \times 10^{-5}$ (Student's $t$-test).

two-hybrid experiment (Y2H, Fig 4A). Immunoprecipitation of Euc1 revealed an up-shifted band that was detected by an anti-SUMO antibody. This band was diminished in *ubc9-1* and lost in *siz1Δ siz2Δ* cells (Fig 4B). Mutation of putative SUMOylation consensus sites identified a single point mutation (*euc1-K231R*, hereafter

*euc1-KR*) that affected Euc1 SUMOylation (Fig 4B). In denaturing NiNTA pull-downs (NiNTA-PDs) of His-tagged SUMO, we detected a band corresponding to monoSUMOylated Euc1 and a weaker band further up-shifted, presumably representing diSUMOylation (Fig 4C). In turn, SUMOylation was strongly reduced in *euc1-KR* cells (Fig 4C), indicating that K231 is the major SUMOylation site in Euc1.

Of note, SUMOylation of Euc1 at K231 is required for the formation of ub-hotspots (Fig 4D). Interestingly, however, Euc1 enrichment itself was also diminished by up to 10-fold in *euc1-KR* cells for two out of three tested sites (Fig EV2A). These data therefore suggest that Euc1 SUMOylation is involved in ub-hotspot function, by regulating Euc1 DNA binding and/or Slx5/Slx8 recruitment.

**SUMOylated Euc1 recruits Slx5/Slx8 to ub-hotspots**

To test whether Euc1 is a ubiquitylation substrate of Slx5/Slx8, we performed denaturing NiNTA-PDs of His-ubiquitin to reveal covalently modified ubiquitylation substrates (Fig 4E). Probing for [3FLAG]Euc1, we detected an up-shifted smear, presumably corresponding to polyubiquitylated Euc1 species (Fig 4E). Notably, Euc1 ubiquitylation was almost entirely lost in a strain harboring a catalytically dead Slx8 RING domain variant (*slx8-CD*, Fig 4E) and also in *slx5Δ* cells (Fig EV2B), indicating STUbL-dependent ubiquitylation. Although steady-state levels of Euc1 were mildly enhanced in *slx5Δ* and SUMOylation-deficient *euc1-KR* cells, both WT Euc1 and Euc1-KR were stable in cycloheximide chase experiments (Fig EV2C–E). We therefore conclude that Slx5/Slx8-catalyzed ubiquitylation either does not lead to Euc1 degradation or that only a small fraction of Euc1 is regulated by Slx5/Slx8.

Euc1 appears to not be the only ubiquitylation substrate at ub-hotspots, as Euc1 signals at ub-hotspots did not correlate with ubiquitylation signals and Euc1 also did not accumulate in *cdc48* mutant strains—in contrast to ubiquitylation—or in ubiquitylation-deficient *slx5Δ* and *slx8Δ* strains (Fig EV2F and G, see also Fig EV2B). We therefore hypothesized that Euc1 and its SUMOylation act as cofactors required for Slx5/Slx8 recruitment. Indeed, when we tested enrichment of Slx5 and Slx8 at ub-hotspot sites in *euc1Δ* or *euc1-KR* strains, both proteins were entirely lost (Fig 4F and G). Consistently, Slx8 recruitment also depended on a functional SUMOylation machinery (Fig EV2H and I). Furthermore, our ChIP data suggested that while Slx5 was recruited in the absence of Slx8 (*3HA-Slx5 slx8Δ*, Fig 4F), Slx8 recruitment to ub-hotspots was Slx5-dependent (*Slx8-9myc slx5Δ*, Fig 4G).

In summary, our data demonstrate that Euc1 plays a major role in regulating ub-hotspots by facilitating Slx5/Slx8 recruitment. However, while Euc1 is a Slx5/Slx8 ubiquitylation substrate, it seems likely that there are additional Slx5/Slx8 substrates bound to ub-hotspots that are cleared from DNA by Cdc48.

**Specific interaction sites mediate SUMO-SIM-independent Euc1-Slx5 binding**

Substrate targeting of STUbLs is thought to rely on multiple SUMO-SIM contacts to confer specificity for polySUMOylated substrates (Sriramachandran & Dohmen, 2014). In contrast, Slx5/Slx8 recruitment to ub-hotspots appears to involve monoSUMOylated Euc1 (Fig 4C). Moreover, ubiquitin enrichment at the ub-hotspots was unchanged in polySUMO chain-defective cells expressing only

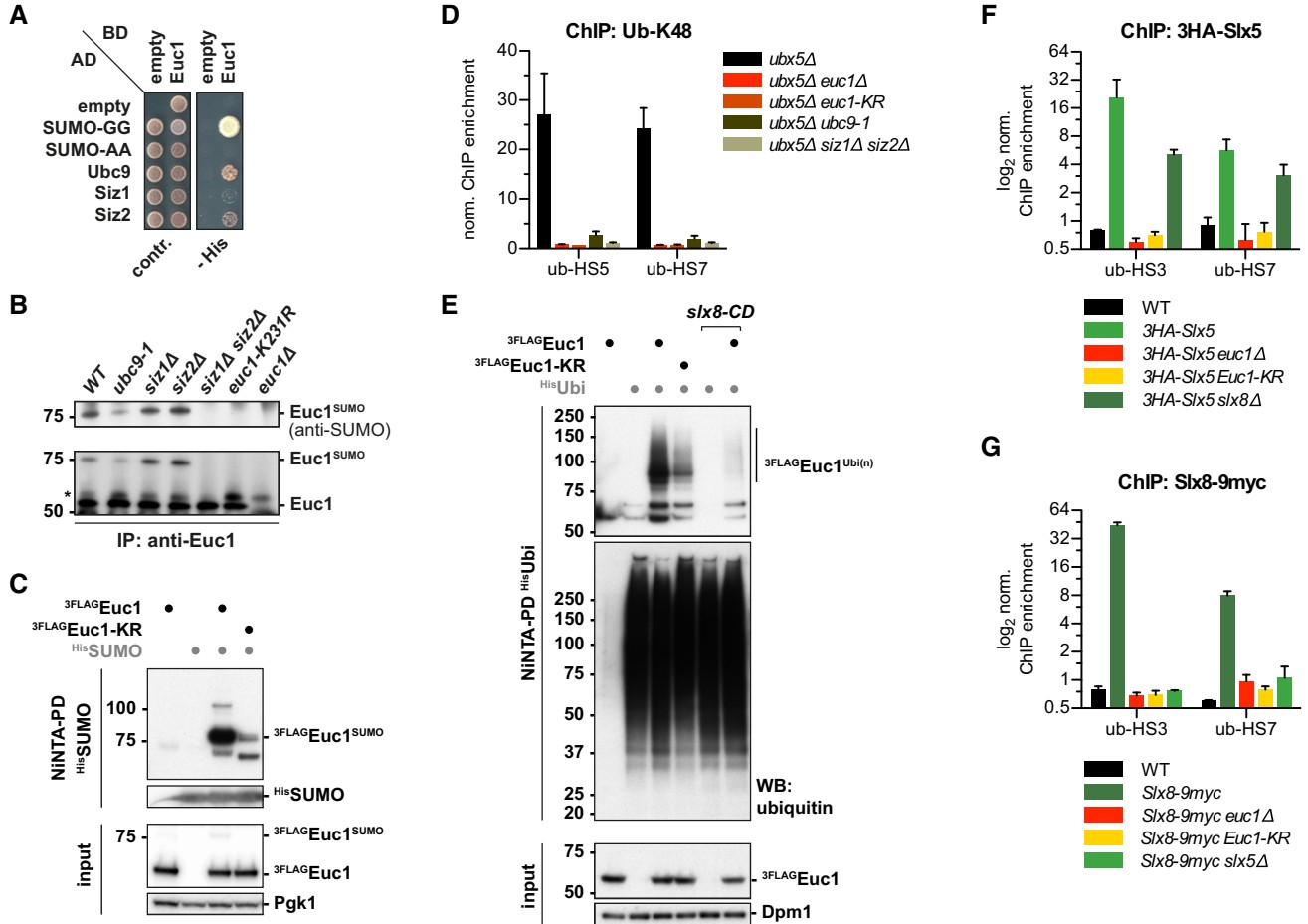

**Figure 4. Euc1 is SUMOylated, recruits Slx5 to ub-hotspots, and is ubiquitylated in a Slx5/Slx8-dependent manner.**

A   Euc1 interacts with SUMO pathway proteins in a yeast two-hybrid (Y2H) assay. A Y2H reporter strain (PJ69-7a) was transformed with Gal4 DNA-binding domain (BD) and Gal4 activation domain (AD) fusion constructs in the indicated combinations. AD-SUMO-GG can be conjugated to SUMOylation substrates, while AD-SUMO-AA is conjugation-deficient. Cells were spotted on control media or selective media (- His) and grown for 3 and 6 days, respectively.

B   Euc1 is SUMOylated by Ubc9, Siz1, and Siz2 on lysine 231. Euc1 was immunoprecipitated from the indicated strains, and eluates were probed by WB against SUMO (top) and Euc1 (bottom). An up-shifted band corresponding to Euc1$^{SUMO}$ was detected with both antibodies. Asterisks denote non-specific bands.

C   Euc1 is predominantly monoSUMOylated on lysine 231. Denaturing NiNTA pull-downs (NiNTA-PD) with strains expressing His-tagged SUMO ($^{His}$SUMO) as indicated and $^{3FLAG}$Euc1 constructs under the control of an *ADH* promoter. Covalently SUMO-modified proteins were enriched and eluates probed with a FLAG antibody to visualize SUMOylated Euc1. Eluates were probed for SUMO to control for equal pull-down, and Pgk1 was probed as input control. Euc1-KR denotes the K231R mutation here and hereafter.

D   Euc1 SUMOylation is required for ub-HS formation. Ub-K48 ChIP quantified by qPCR for the indicated strains. The *ubx5Δ* background was used to enhance the ubiquitin signal, and similar results were obtained in a WT background. Data represent means ± SD (n = 3).

E   Euc1 ubiquitylation depends on Slx8 and partly on Euc1 SUMOylation. Denaturing NiNTA-PDs as described for (C) but with strains expressing His-ubiquitin ($^{His}$Ubi) and $^{3FLAG}$Euc1 under the control of the *ADH* promoter. WBs were developed with a FLAG antibody to probe for $^{3FLAG}$Euc1, a ubiquitin blot served as PD-control and Dpm1 as input control. The *slx8-CD* allele carries the C206S and C209S mutations (Xie *et al*, 2007).

F, G   SUMOylated Euc1 is required to recruit Slx5 and Slx8 to ub-HSs. ChIP against Slx5 and Slx8 in the indicated genetic backgrounds was quantified by qPCR. Note that Slx5 is still recruited in the absence of Slx8 (*3HA-Slx5 slx8Δ*), but not vice versa (*Slx8-9myc slx5Δ*). Data for WT and 3HA-Slx5 in (F) and WT and Slx8-9myc in (G) are from Fig 1D and are shown here for comparison. Data represent means ± SD (n = 2).

Source data are available online for this figure.

lysine-free SUMO (*SUMO-KRall*) (Fig EV3A). What then underlies the specificity of Slx5/Slx8 recruitment to ub-hotspots, given the high prevalence of SUMOylation on chromatin (Chymkowitch *et al*, 2015)?

Our ChIP data suggested that SUMOylated Euc1 interacts with Slx5 to recruit the Slx5/Slx8 complex (Fig 4F), and we could confirm an annotated interaction between Euc1 and Slx5 in Y2H

experiments (Figs 5A and EV3B, Appendix Fig S5A and B) (Yu *et al*, 2008). The Euc1-Slx5 interaction was further enhanced when we used a Slx5 fragment lacking the C-terminal RING domain (Slx5-RINGΔ). Interestingly, however, Slx5-RINGΔ still interacted with an Euc1-KR construct lacking the SUMO target site (Fig EV3B). Moreover, recombinant, non-SUMOylated Euc1 co-precipitated Slx5 *in vitro* (Fig 5B), indicating that Euc1 SUMOylation is not strictly

required for interaction. We mapped a minimal interacting fragment to aa 81–183 of Euc1 (Figs EV3B and 5A), a stretch lacking the SUMOylation site, but comprising the CC domain, which is crucial for Euc1 dimerization (Appendix Fig S5C and D). Also, co-immunoprecipitation (co-IP) experiments with [3FLAG]Euc1 showed a robust *in vivo* interaction with Slx5 that was independent of Euc1 SUMOylation at K231 (Fig 5C, lanes 2–3). Overall, this indicates additional, SUMO-independent Euc1-Slx5 interaction sites. Recently, binding of Matα2 to DNA has been shown to be required for Slx5/Slx8-mediated degradation (Hickey *et al*, 2018). Euc1, however, was still able to co-precipitate Slx5 when its DNA-binding domain was mutated (Euc1-DBD*, Fig 5C, lane 4).

Truncation of the Euc1 CC domain abolished the interaction with Slx5 (Fig EV3B) as well as Euc1 dimerization in Y2H (Appendix Fig S5C and D); however, Slx5 interaction was also lost by truncation of the region between aa 140 and 183, indicating a potential Slx5-binding site in this region (compare Fig EV3B and Appendix Fig S5C, Euc1 1–140). To guide our search, we used HH-MOTiF for *de novo* motif prediction (Prytuliak *et al*, 2017) using Euc1 aa 81–183 and a set of putative Slx5 substrates as query. We introduced mutations in predicted binding sites downstream of the CC domain: Two of these strongly diminished binding to Slx5 while leaving dimerization intact (Slx5-binding mutant 1 and 2, SBM1: aa 139–143 ENQKK>ANAAA, SBM2: aa 162–165 KEVF>AAAA, Fig 5D and Appendix Fig S6A and B). Notably, both mutations strongly reduced the interaction between Euc1 and Slx5 in co-IP experiments (Fig 5E, lanes 2–5), while additional mutation of the Euc1 SUMOylation site had little effect (KR mutation, Fig 5E, lanes 6–9).

We also mapped the region of Slx5 that mediates Euc1 binding to a previously uncharacterized region between aa 200 and 335 (Fig EV3C). We designate this region, which does not contain any predicted SIMs, Slx5 middle domain (Slx5-Md). Slx5-Md is required and sufficient for interaction with the Euc[81–183] fragment (Fig 5F) and Slx5-binding-deficient mutants of Euc1 strongly diminished the interaction (Fig 5G). We conclude that interaction with Euc1 involves the Slx5-Md and that SUMO-SIM contacts are not strictly required, but likely contributing, suggesting bipartite substrate recognition (Fig 5H).

## Specific Euc1-Slx5 interaction sites are required for ub-hotspots

We tested whether ubiquitylation of Euc1 or ub-hotspot formation would be affected by mutations that specifically abolish SUMO-independent contacts between Euc1 and Slx5. Using NiNTA-PDs with His-ubiquitin, we found that both Euc1 Slx5-binding mutants (SBM1 and SBM2), as well as the double mutant, showed a reduction in Euc1 ubiquitylation (Fig 6A, lanes 3–6). When we additionally mutated the Euc1 SUMOylation site (KR), Euc1 ubiquitylation was further decreased (Fig 6A, lanes 7–10). Furthermore, the WT Slx5-construct complemented the loss of ubiquitylation in an *slx5Δ* strain (Fig 6B, lane 2), whereas expression of the Slx5-SIM* construct with all SIMs mutated yielded reduced Euc1 ubiquitylation, and expression of the Slx5-MdΔ construct abolished Euc1 ubiquitylation (Fig 6B, lanes 3–4).

The Slx5-Md (aa 201–335) overlaps with a stretch that has been implicated in nuclear localization, Slx8 and Slx5 interaction (Westerbeck *et al*, 2014), as well as an auto-ubiquitylation-protective "lysine desert" (Sharma *et al*, 2017). To exclude hypomorphic

effects, we first tested the ability of the *SLX5* alleles to complement hypersensitivity of *slx5Δ* cells on replication stress induced by hydroxyurea (HU) (Xie *et al*, 2007). The *slx5-MdΔ* allele complemented the growth phenotype on HU, while the *slx5-SIM** did not (Appendix Fig S7A–C). Complementation was not influenced by an N-terminally fused NLS. Second, we monitored turnover of a fragment of the known Slx5/Slx8-substrate Matα2 (Hickey & Hochstrasser, 2015), which was not affected in *slx5-MdΔ* cells, suggesting that the involvement of the Md is substrate-specific (Appendix Fig S7D).

Last, we tested whether the ubiquitin signal at ub-hotspots was influenced and found it to be reduced in cells that expressed Euc1-SBM1 as only copy of Euc1, while it was completely lost in *euc1-SBM2* and *euc1-SBM1+2* cells (Fig 6C, top). As expected, the recruitment of Slx5 to ub-hotspots was also diminished for *euc1-SBM1* cells and lost for *euc1-SBM2* and *euc1-SBM1+2* cells (Fig 6C, bottom), as was Euc1 binding (Appendix Fig S7E–G), suggesting mutually dependent binding of Euc1 and Slx5/Slx8 to ub-hotspots. Conversely, we also tested whether the Slx5-Md and SIMs would be required for ubiquitylation at the ub-hotspots. When we complemented an *slx5Δ ubx5Δ* strain with plasmids encoding Slx5 variants, we observed that WT *SLX5* was able to restore ub-hotspots, while the *slx5-SIM** and the *slx5-MdΔ* alleles were not (Fig 6D, left). Similar to Euc1-SBM1/2, mutation of Slx5-SIMs or deletion of Slx5-Md led to loss of Slx5 recruitment (Fig 6D, right), and to strongly diminished recruitment of Euc1 itself (Appendix Fig S7H), indicating the presence of an Euc1-Slx5/Slx8 complex at ub-hotspots.

Overall, our data point toward a bipartite or multivalent interaction between Euc1 and Slx5/Slx8 where not only Slx5-SIMs and Euc1 SUMOylation, but also substrate-specific contacts between Euc1 and Slx5 are critical for Euc1 ubiquitylation and the formation of ub-hotspots on chromatin (summarized in Fig 6E).

## The Slx5/Slx8-dependent ub-hotspot pathway controls Euc1 function

The sophisticated STUbL recruitment mechanism suggested that Slx5/Slx8 could regulate Euc1 and ub-hotspot function. To determine whether Euc1 might be active as a transcriptional activator at ub-hotspots, we performed transcriptome analysis in *euc1Δ*, *euc1-KR* cells and in cells overexpressing *EUC1* (pGAL-EUC1). In *euc1Δ* cells, only 2 of 15 genes in direct proximity of ub-hotspots were significantly up- or downregulated in expression, with opposing trends (*RCO1* up, *SSF2* down, Fig 7A, Appendix Fig S8A). More than 150 genes showed a significant change in *euc1Δ* ($P < 0.01$), suggesting that Euc1 may play a widespread function in the regulation of gene expression or reflecting a cellular adaptation to loss of *EUC1* (Dataset EV6). Gene ontology (GO) term enrichments suggest an upregulation of small molecule metabolic processes (e.g., carboxylic acid and amino acid metabolism, Dataset EV6). Overall, these data do therefore not provide support for a role of Euc1 as direct transcriptional activator of ub-hotspot adjacent genes, at least not in the tested conditions.

We noticed that mutations affecting Euc1 SUMOylation (*euc1-KR*, *siz1Δ siz2Δ*) led to stronger reporter gene activation using Euc1-N-Gal4-BD reporter constructs (Appendix Fig S8B and C). Importantly, however, we failed to detect strongly deregulated genes in a

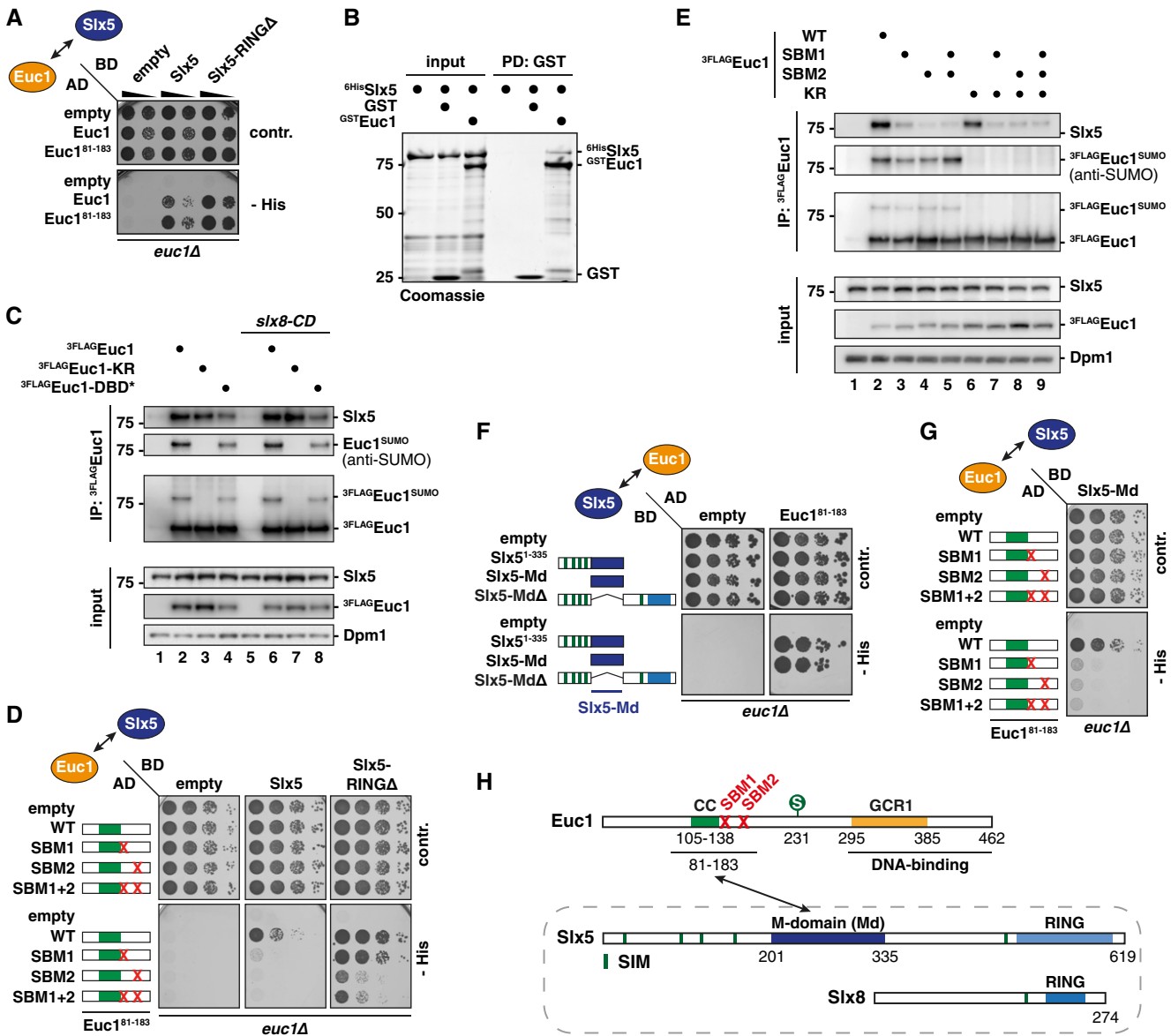

**Figure 5. Specific substrate–ligase interaction sites mediate SUMO-SIM-independent Euc1-Slx5 binding.**

A   Euc1[81–183] binds to Slx5 in Y2H assays. Y2H assay performed as in Fig 4A. Serial dilutions were spotted for each plasmid combination, and cells were grown at 30°C for 2 days. The Slx5-RINGΔ construct is deleted for the Slx5 C-terminus from the beginning of the RING domain (Δaa 488–619). See also Fig EV3B.

B   Euc1 binds to Slx5 *in vitro*. Recombinant purified GST or [GST]Euc1 was used in GST pull-down assays to co-precipitate recombinant [6His]Slx5.

C   Euc1 binds to Slx5 *in vivo*. Cell lysates from an *euc1Δ* strain expressing the indicated [3FLAG]Euc1 constructs from plasmids (under *EUC1* promoter) were subjected to immunoprecipitation using anti-FLAG beads. IP eluates were probed by WB with Slx5, SUMO, and Euc1 antibodies, and inputs were probed with Slx5, FLAG, and Dpm1 antibodies (top to bottom). Note that the Euc1-Slx5 interaction is independent of the Slx8 ligase activity (*slx8-CD*, lanes 5–8).

D   Euc1 Slx5-binding mutants (SBM1, SBM2) affect Euc1-Slx5 interaction in Y2H assays. AD-Euc1[81–183] constructs harboring mutations in the region required for Slx5 binding were probed for interaction with BD-Slx5 constructs as described in Fig 4A. Mutations: SBM1: aa 139–143 ENQKK>ANAAA, SBM2: aa 162–165 KEVF>AAAA. Serial dilutions were spotted, and cells were grown at 30°C for 2 days. See Appendix Fig S6A for expression levels.

E   Euc1-SBM constructs show defective Slx5 binding *in vivo*. Mutations described in (D) were introduced into full-length [3FLAG]Euc1 constructs (with or without the K231R mutation), and FLAG IPs were performed as described in (C). WB analysis showed strong Slx5-binding defects for the SBM1/SBM2 and SBM1+2 constructs (top panel, immunoprecipitated Slx5). WBs were probed as in (C).

F   The Slx5 middle domain (Slx5-Md) is required for interaction with Euc1. Y2H assays with AD-Euc1[81–183] and BD-Slx5 constructs. Slx5-Md: aa 201–335, Slx5-MdΔ: Δaa 201–338. Serial dilutions were spotted, and cells were grown at 30°C for 4 days. See also Fig EV3C.

G   Euc1-SBM constructs show defective binding to the Slx5-Md. Y2H assay with the indicated constructs as in (D). See Appendix Fig S6A for expression levels.

H   Schematic representation of Euc1, Slx5, and Slx8, domain features, and interactions. Domains and protein lengths are drawn to scale, and numbers below each bar denote amino acid positions. The mapped interaction between Euc1 and Slx5 is indicated by an arrow. CC: coiled-coil domain, SBM: Slx5-binding mutant, S: SUMO, SIM: SUMO-interacting motif.

Source data are available online for this figure.

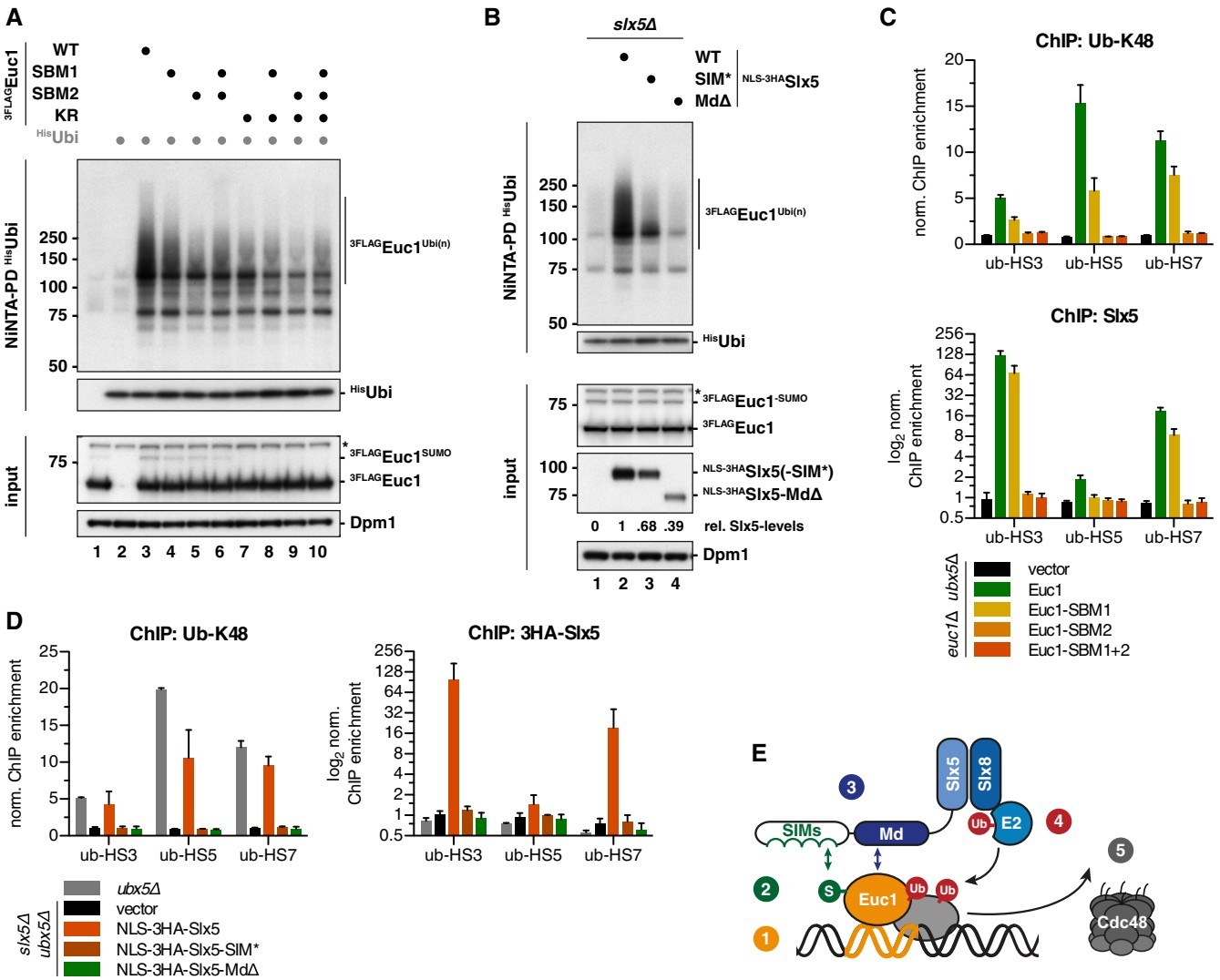

**Figure 6. Specific Euc1-Slx5 interactions are required for ub-hotspots and Euc1 ubiquitylation.**

A   Euc1 Slx5-binding mutations impair Euc1 ubiquitylation. His-ubiquitin-modified proteins were enriched by denaturing NiNTA-PDs as described in Fig 4E. 3FLAG-tagged Euc1 constructs were expressed from plasmids under the control of the *ADH* promoter. NiNTA-PD eluates were probed with FLAG and ubiquitin (P4D1) antibodies. Whole cell lysates (input) were probed with FLAG and Dpm1 antibodies. Asterisk denotes a non-specific band.

B   The Slx5-SIMs and Slx5-Md are required for Euc1 ubiquitylation. NiNTA-PDs for His-ubiquitin as in (A). All strains expressed wild-type ³FLAGEuc1 from plasmids under the *ADH* promoter and His-ubiquitin. Slx5 constructs were expressed from plasmids under the control of the endogenous promoter. WBs were probed as described in (A), and Slx5-levels were probed using an HA antibody. Slx5-SIM*: SIMs 1–4 were mutated as described (Xie *et al*, 2010); for SIM5, aa 477–479 (IIV) were mutated to alanines. To avoid possible mislocalization by deletion of the Slx5-Md, which overlaps with a putative NLS (Westerbeck *et al*, 2014), an N-terminal NLS was fused to all constructs. See Appendix Fig S7C for HU complementation.

C   Euc1 Slx5-binding mutations (SBM1/SBM2) reduce/abolish ub-HSs and recruitment of Slx5 to ub-HSs. ChIP-qPCR for ub-K48 (top) or Slx5 (using Slx5 antibody, bottom) in *euc1Δ ubx5Δ* cells expressing the indicated constructs from plasmids under the control of the *EUC1* promoter. Data represent means ± SD (n = 3). See also Appendix Fig S7E–G.

D   Slx5-SIMs and the Slx5-Md are required for the formation of ub-HSs and recruitment of Slx5. ChIP-qPCR for ub-K48 (left) or Slx5 (anti-HA ChIP, right) in *ubx5Δ* cells or *slx5Δ ubx5Δ* cells complemented with plasmids expressing the indicated Slx5 constructs under the control of the *SLX5* promoter. Data represent means ± SD (n = 2). See also Appendix Fig S7H.

E   Schematic model for the proposed sequence of events at ub-hotspots. See main text (Discussion) for details. SIM: SUMO-interacting motif, Md: middle domain, S: SUMO, Ub: ubiquitin.

Source data are available online for this figure.

transcriptome analysis of *euc1-KR* cells (Appendix Fig S8D, Dataset EV7) putting further in question whether Euc1 acts as a transcription factor in cells.

Supporting this notion, none of the ub-hotspot adjacent genes were strongly deregulated upon *EUC1* overexpression (Fig 7B, Appendix Fig S8E, Dataset EV8). Four of them showed mild

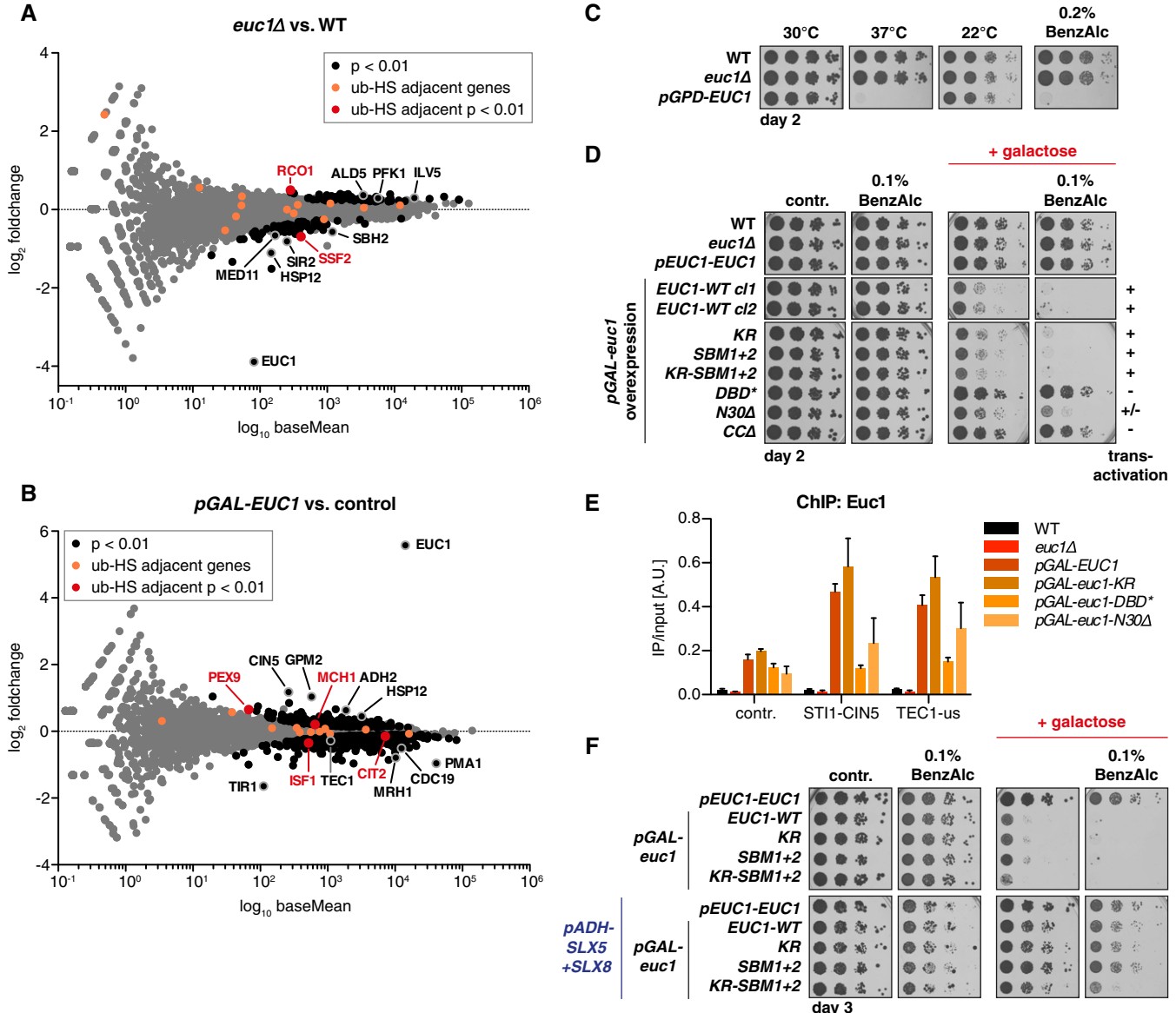

**Figure 7. The ub-hotspot pathway is required to regulate aberrant Euc1 function.**

A Deletion of *EUC1* does not lead to ub-HS adjacent gene deregulation, but rather widespread transcriptome changes. Total RNA isolated from WT and *euc1Δ* cells was polyA-enriched and sequenced (RNAseq, triplicates). Significance testing was based on the Wald test, see Materials and Methods for details. See also Appendix Fig S8A for quantification of selected transcripts by qPCR. baseMean: mean expression levels across all samples.

B RNAseq transcriptome analysis of *EUC1* overexpression as in (A). *Δeuc1* cells with *pGAL-EUC1* or *pEUC1-EUC1* (control) integrated at the *URA3* locus (YIplac211) were grown to mid-log phase, and 2% galactose was added for 3 h. See also Appendix Fig S8E.

C *EUC1* overexpression is toxic at elevated temperatures and upon exposure to the membrane-fluidizing drug benzyl alcohol (BenzAlc). For (C, D, F), serial dilutions of the indicated strains were spotted and grown on YPD control (or selective media) plates or under conditions as indicated. See also Fig EV4A and B.

D *EUC1* overexpression toxicity depends on DNA binding and transactivation. The indicated *EUC1* alleles with endogenous or galactose-inducible promoters were integrated at the *URA3* locus (YIplac211, *euc1Δ* background) and spotted on glucose control or galactose plates to induce *EUC1* overexpression. See also Appendix Fig S9A–C.

E Aberrant binding of overexpressed *EUC1* partially depends on Euc1 DNA binding. ChIP-qPCR of Euc1 after 3-h galactose induction. Note that IP/input ratios of Euc1 signals are shown, also for the control locus (contr., *TOS1* promoter) to highlight Euc1 binding at non-ub-HS sites. *STI1-CIN5*: intergenic region. *TEC1-us*: upstream (promoter) region. Data represent means ± SD (*n* = 3). A.U.: arbitrary units. See also Fig EV4C and D.

F Simultaneous overexpression of *SLX5* and *SLX8* rescue *EUC1* toxicity. Experiment as in (D), but with concomitant plasmid-borne overexpression (*ADH* promoter) of *SLX5* and *SLX8* (bottom panels) and on media selecting for *SLX5*/*SLX8* plasmids. See also Fig EV4E and Appendix Fig S9D.

up- or downregulation, but many other genes showed stronger expression changes, in particular genes involved in cellular metabolism. Interestingly, however, overexpression of *EUC1* (*pGPD-EUC1*, *pGAL-EUC1*) led to lethality at elevated temperatures or upon exposure to the membrane-fluidizing drug benzyl alcohol (Lone *et al*, 2015), while deletion of *EUC1* did not impair growth under

these conditions (Figs 7C and EV4A). In line with a strict control by Slx5/Slx8, overexpressed *EUC1* also led to a strong growth phenotype when paired with *slx5Δ* or *slx8Δ* (Fig EV4B). Overexpression toxicity was not influenced by mutations that abolish ub-hotspot formation, such as SUMOylation-deficient *euc1-KR* or Slx5-binding-deficient *euc1-SBM1+2* (Fig 7D, Appendix Fig S9A), suggesting that these phenotypes are not caused by over-active ub-hotspots. In contrast, overexpression toxicity was dependent on DNA binding and at least in part on the N-terminal transactivation domain, as well as on the CC domain, that also is essential for transactivation (*DBD**, *N30Δ*, *CCΔ* alleles, Fig 7D, Appendix Fig S9A and B). This effect was not due to impaired nuclear localization, as Euc1-WT, Euc1-KR, and Euc1-DBD* all localized to the nucleus (Appendix Fig S9C). Compared to endogenous Euc1, the overexpressed protein bound to additional loci as tested in ChIP-chip and confirmed in ChIP-qPCR (*STI1-CIN5* intergenic, *TEC1* upstream (us), Figs 7E and EV4C), and this additional binding was strongly reduced in *euc1-DBD** and *euc1-N30Δ* cells. Importantly, binding to these additional loci did not lead to the formation of additional ub-hotspots, collectively indicating that effects of Euc1 overexpression appear unrelated to ub-hotspots (Figs 7E and EV4D).

Widespread deregulation of genes upon *EUC1* overexpression could reflect direct Euc1-dependent transcriptional regulation of these genes, but indirect effects are possible as well, given that *EUC1* overexpression deregulated other transcription factors, such as *CIN5* (a TF mediating pleiotropic drug resistance and salt tolerance) and *TEC1* (a TF targeting filamentation genes) (Fig 7B, Appendix Fig S8E, Dataset EV8).

Importantly, overexpression of *SLX5/SLX8* (*pADH-SLX5+SLX8*) rescued Euc1 toxicity and prevented aberrant binding at *STI1-CIN5* and *TEC1-us* loci (Figs 7F and EV4E), but overexpression of the SUMOylation- and Slx5-binding-deficient *euc1-KR-SBM1+2* could not be compensated by additional *SLX5/SLX8* overexpression (Figs 7F and EV4E, Appendix Fig S9D). Taken together, we conclude that Euc1 needs to be tightly controlled by the STUbL pathway in order to restrict its action to sites of ub-hotspots and that spurious Euc1 leads to changes in gene expression, likely via its transactivation function. In contrast, endogenous Euc1 seems to have no or only a limited effect on transcription of ub-hotspot adjacent genes.

### *EUC1* shows genetic interactions with regulators of gene expression upon thermostress

To explore the function of Euc1 and ub-hotspots, we investigated genetic interactors of *EUC1*. Previous high-throughput studies have suggested candidate genetic interactors that show GO term enrichments for chromatin organization, negative regulation of transcription, histone deacetylation, and related processes (Dataset EV9; Zheng *et al*, 2010; Costanzo *et al*, 2010, 2016). Indeed, we observed a negative genetic interaction of *EUC1* with *HTZ1*, encoding for histone H2A.Z and the functionally linked nucleosome remodeling complex SWR1-C (*SWR1* and *YAF9* genes), in particular upon thermostress (heat or cold), DMSO, or HU treatment (Fig 8A, Appendix Fig S10A and B). Histone H2A.Z deposition by SWR1-C has been implicated in the regulation of inducible promoters, heterochromatin maintenance, and genome maintenance (Billon & Côté, 2013). Genetic studies previously revealed sensitivity

and interactions of mutants in the H2A.Z-SWR1-C pathway upon heat and DMSO stress, and H2A.Z was proposed as a nucleosomal "thermosensor" in *A. thaliana* and budding yeast (Zhang *et al*, 2005; Lindstrom *et al*, 2006; Kumar & Wigge, 2010; Gaytán *et al*, 2013). The precise molecular basis for DMSO toxicity is unclear. However, interference with membrane organization and inhibition of histone deacetylases (HDACs) have been described (Gurtovenko & Anwar, 2007; Marks & Breslow, 2007). Other negative genetic interactions became apparent with *NPL3*, an mRNA splicing and processing factor, and *STB5*, a transcription factor involved in oxidative and multidrug stress responses (Fig 8A, Appendix Fig S10A and C–F).

Most strikingly, deletion of *EUC1* led to a pronounced aggravation of the described heat sensitivity of cells deficient in the Rpd3L histone deacetylase complex (*dep1Δ*, *rxt2Δ*, *sds3Δ*, Fig 8B and C) (Ruiz-Roig *et al*, 2010). Rpd3 is an orthologue of human class I HDACs and is part of two major HDAC complexes in yeast (Carrozza *et al*, 2005): (i) Rpd3L, which is primarily recruited to promoter regions for transcriptional repression, but also activation, e.g., upon heat stress; and (ii) Rpd3S, which is recruited to transcribed regions by Set2-mediated histone H3K36 methylation and has been described to control cryptic intragenic transcription by deacetylating histones within coding regions (de Nadal *et al*, 2004; Carrozza *et al*, 2005; Ruiz-Roig *et al*, 2010). We note that the pronounced heat and DMSO sensitivity of shared Rpd3L and Rpd3S subunit mutants (*rpd3Δ*, *sin3Δ*) was not further increased by *euc1Δ* (Appendix Fig S10G and H) and that phenotypes of Rpd3S subunit mutants (*rco1Δ*, *eaf3Δ*) were not further increased by *euc1Δ* (Fig 8D). Interestingly, combination of mutations in Rpd3S (*rco1Δ*, *eaf3Δ*) and Rpd3L (*sds3Δ*) showed strongly increased sensitivity to heat stress and DMSO, suggesting functional redundancy of the Rpd3L and Rpd3S complexes (Fig 8D). In this background, the effect of Rpd3S mutations was not further aggravated by deletion of *EUC1*, indicating epistasis and that Euc1 acts with Rpd3S in a pathway that mediates thermotolerance and resistance to DMSO stress (Fig 8D and E). In agreement with our assignment of *EUC1* to an Rpd3S-related function, we found significant correlations of genes downregulated in *euc1Δ* with those downregulated in published *rpd3Δ* transcriptomes, but also with datasets for *set2Δ*, which acts as an upstream regulator in the Rpd3S pathway (Appendix Fig S11A) (McKnight *et al*, 2015; McDaniel *et al*, 2017).

### Euc1-mediated ub-hotspots are crucial during stress responses when gene expression control is impaired

To corroborate a potential role of Euc1 in the response to heat stress, we performed ChIP in cells shifted to 37°C and observed a significant increase of Euc1 recruitment to ub-hotspots, in particular in *sds3Δ* cells (Fig 9A, Appendix Fig S11B). Similar trends were observed in *htz1Δ* and *npl3Δ* cells (Appendix Fig S11C–E). Notably, ubiquitin signals decreased at 37°C, but this effect was also seen for an Euc1-independent ubiquitin-bound region (ub-only-site1), possibly reflecting a decrease in available free ubiquitin at elevated temperatures (Fig 9A) (Finley *et al*, 1987). Overall, these data support a model whereby Euc1- and Slx5/Slx8-dependent ub-hotspots serve a role in gene expression control, which becomes critical upon exposure to thermo- and other stress conditions.

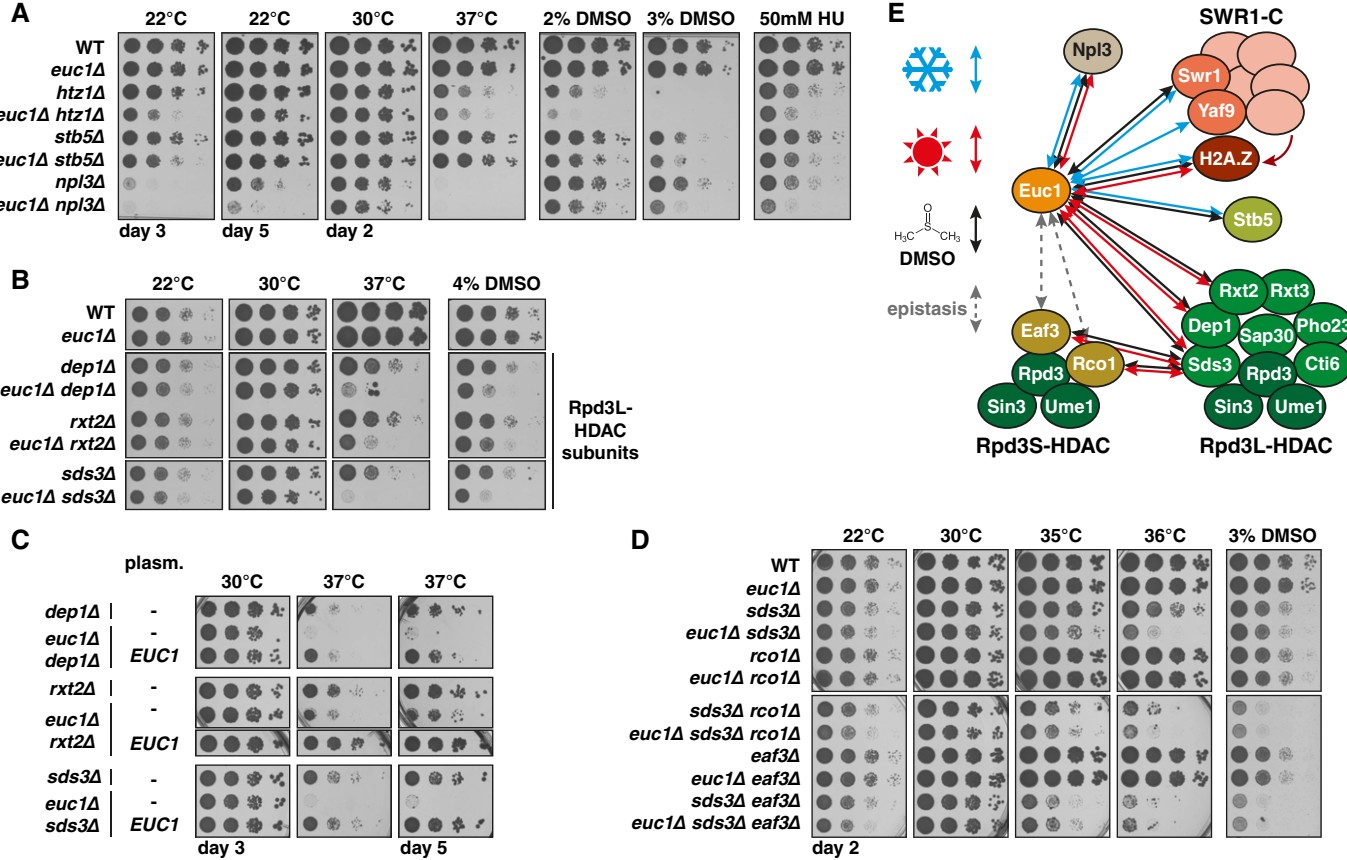

**Figure 8. *EUC1* shows genetic interactions with regulators of gene expression upon thermo- and DMSO stress.**

A, B  *EUC1* displays negative genetic interactions with genes involved in general and specific transcriptional regulation (A), and in particular with members of the Rpd3L histone deacetylase (HDAC) complex (B). For (A–D), serial dilutions of the indicated strains were spotted and grown on YPD control or selective media plates, or under conditions as indicated. For (B), plates were incubated for 2 days (22°C, 30°C, 4% DMSO) or 3 days (37°C). Note that all strains used in (A–D) contained an extra copy of *MED11*, see Appendix Fig S10C–F for details.

C  Plasmid-borne *EUC1* complements genetic interactions with Rpd3L subunits. Empty vector (–) or a plasmid encoding *EUC1* with its endogenous promoter were transformed in the indicated strains and spotted on selective media.

D  *EUC1* and Rpd3S act in a common pathway which is redundant with Rpd3L. *rco1Δ* and *eaf3Δ* (Rpd3S) show similar phenotypes when paired with *sds3Δ* (Rpd3L) as are *euc1Δ* and are epistatic with *euc1Δ*.

E  Graphic summary of *EUC1* genetic interactions as tested in (A–D) and Appendix Fig S10. Arrows indicate negative genetic interactions upon cold (blue), heat (red), or DMSO stress (black), and dashed gray arrows indicate epistatic relationships.

Therefore, we wondered whether transactivation or ub-hotspot functions would be required in the context of endogenous Euc1 function. Plasmid-borne expression of Euc1 was used to complement *euc1Δ* phenotypes. Notably, transactivation-deficient *euc1-N30Δ* and WT *EUC1* could both rescue temperature and DMSO sensitivity in *sds3Δ* and *npl3Δ* backgrounds (Fig 9B, Appendix Fig S12A). In contrast, *euc1* mutants deficient in ub-hotspot formation (*KR, SBM1+2, KR-SBM1+2, DBD*, CCΔ,* Figs 4D and 6C, Appendix Figs S3E and S12B) behaved like *euc1Δ*, suggesting that specifically ub-hotspots are required for thermotolerance and resistance against DMSO in the context of the Rpd3S pathway (Fig 9B, Appendix Fig S12A). Consistently, we found that deletion of *SLX5* and *SLX8* enhanced *sds3Δ* heat sensitivity as well (Fig 9C, Appendix Fig S12C) and that deletion of *EUC1* did not further aggravate these phenotypes. Also, these genetic interactions are likely dependent on ub-hotspot function, as they could be rescued by plasmid-borne expression of WT *SLX5* but only

to a minor extent by *slx5-SIM** and *slx5-MdΔ* (Appendix Fig S12D).

We therefore conclude that Euc1- and Slx5/Slx8-dependent ub-hotspots are critical for thermotolerance and the response to other stresses, and that Euc1 is functionally linked to gene regulation by the Rpd3S histone deacetylase.

## Discussion

### A complex cascade controls proteins bound at ub-hotspots

Chromatin is a tightly controlled cellular environment. In recent years, STUbLs and the Cdc48/p97 segregase have emerged as components of the ubiquitin–proteasome system that critically control protein–DNA transactions across species (Ramadan *et al*, 2007; Wilcox & Laney, 2009; Ndoja *et al*, 2014; Maric et al, 2014;

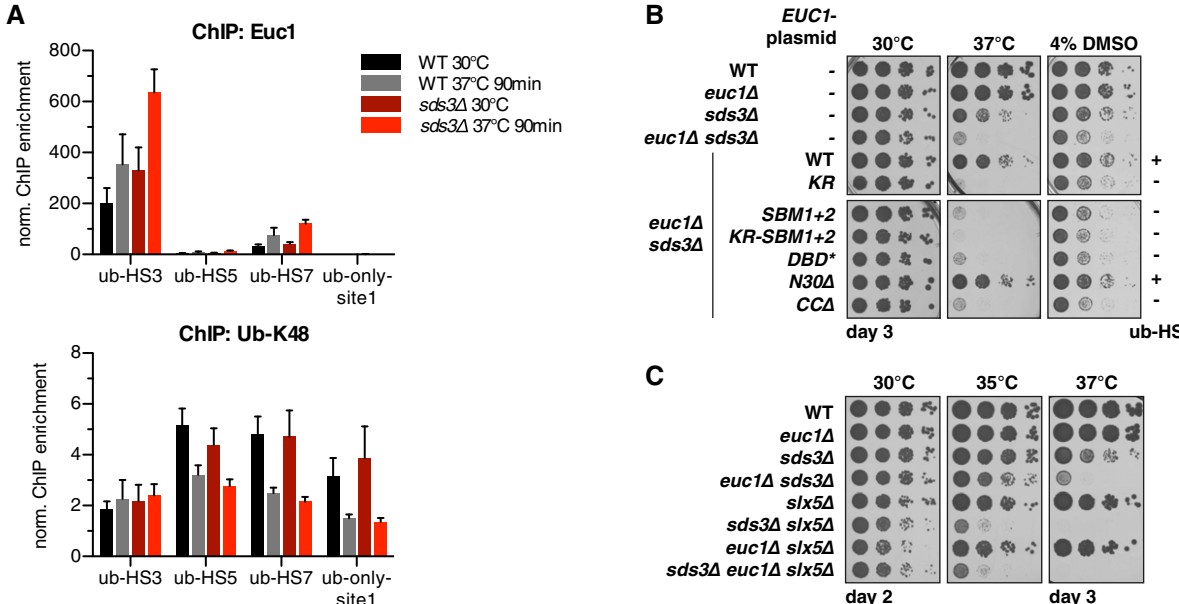

**Figure 9. Euc1-mediated ub-hotspots and Slx5/Slx8 are crucial for a functional stress response under impaired gene expression control.**

A Euc1 is recruited to ub-HSs upon mild heat stress (37°C, 90 min). ChIP-qPCR experiments with strains and conditions as indicated. Data represent means ± SD (n = 4), see Appendix Fig S11B for statistical analysis.

B The ability to form ub-hotspots is crucial for endogenous *EUC1* function. The genetic interaction of *EUC1* with *SDS3* (Rpd3L complex) was rescued with plasmid-borne *EUC1* alleles, and serial dilutions were spotted on selective media. (−) denotes empty vector. See also Appendix Fig S12A and B.

C *SLX5* also shows negative genetic interactions with *SDS3* upon heat stress. Serial dilutions of the indicated strains were spotted on YPD and incubated as indicated. See also Appendix Fig S12C and D. Note that all strains used in (A–C) contained an extra copy of *MED11*, see Appendix Fig S10C–F for details.

Franz *et al*, 2016; Nie & Boddy, 2016). By generating genome-wide ubiquitin and Slx8 binding profiles, in this work we identified specific intergenic sites in the yeast genome—ub-hotspots—where the SUMO and ubiquitin pathways converge to control the abundance of proteins on DNA.

We elucidated a cascade of events controlling ub-hotspots suggesting the following model (summarized in Fig 6E): (i) The DNA-encoded ub-HS-motif is bound by Euc1 via its GCR1 domain. (ii) Ubc9, Siz1, or Siz2 modify Euc1 with SUMO, and SUMOylation stabilizes DNA binding of Euc1. (iii) Euc1 recruits Slx5/Slx8 via specific contacts between Euc1 and the middle domain of Slx5 (Slx5-Md), as well as an additional SUMO-SIM-mediated interaction. (iv) Slx5/Slx8 ubiquitylate Euc1 and presumably other targets at ub-hotspots, and (v) the Cdc48$^{Ufd1-Npl4}$ complex, together with Ubx4 and Ubx5, removes K48-linked ubiquitylated proteins from chromatin. Such extracted, ubiquitylated proteins could be either degraded by the proteasome or recycled by deubiquitylation as it has been found for other TFs (Inui *et al*, 2011), even if marked with K48-linked ubiquitin chains (Flick *et al*, 2004). Euc1 itself does not seem to underlie extensive turnover, as it is a very stable protein (Fig EV2E).

Sequence-specific binding and the resulting highly localized ChIP signals offer an attractive explanation as to why signals at ub-hotspots appear so strongly enriched over other events of STUbL binding and ubiquitylation, which are known to occur on chromatin. Importantly, ub-hotspots exhibit a considerable variability in the quantity of Euc1, Slx5/Slx8, and ubiquitin enrichment (Fig 1B, D and E), with an apparent correlation of Euc1 and Slx5/Slx8

enrichment, but not ubiquitin enrichment (Fig EV2F). This phenomenon could be explained by either variable efficiency of ubiquitylation at different ub-hotspots or the association of other proteins with the Euc1 complex. These proteins might then become targets of Slx5/Slx8-dependent ubiquitylation. In line with such an "in trans ubiquitylation" model, other studies have found that STUbLs can also target interaction partners of SUMOylated proteins (Abed *et al*, 2011; Schweiggert *et al*, 2016). Notably, this model implies that ubiquitylation substrates may not necessarily be the same for all ub-hotspots. Overall, such a cascade of binding and modification events at the ub-hotspots offers ample possibility for regulation and fine-tuning.

**Euc1 and ub-hotspots function in tolerance to cellular stress**

What could be the cellular function of Euc1? Euc1 shares similarity in domain architecture with transcription factors (TFs) of the GCR1 domain family, including DNA-binding, transactivation, and coiled-coil dimerization domains (Fig 3A) (Holland *et al*, 1987; Hohmann, 2002). All three previously characterized GCR1 domain TFs function as transcriptional regulators during the adaptation of cells to changing environmental conditions, such as glucose availability and osmotic stress (reviewed in Hohmann, 2002). Notably, while we find that Euc1 can activate transcription in reporter gene assays, our evidence so far does not suggest that Euc1 would function as transcription factor at ub-hotspots. It is possible that Euc1's role as transcription factor is only activated upon a specific, currently unknown stimulus, but we favor a model whereby Euc1 exerts its major

function through the formation of ub-hotspots, which seems independent of transactivation (Fig 9B, Appendix Fig S12A).

Our data highlight negative genetic interactions with other players regulating gene expression (H2A.Z-SWR1-C pathway, *NPL3*, *STB5,* and Rpd3L complex), in particular under stress conditions such as exposure to cold, heat, or DMSO. Consistently, several of the *EUC1* interactors have well-described functions in stress adaptation, apparently by their widespread functions in gene expression (de Nadal *et al*, 2004; Kumar & Wigge, 2010; Ruiz-Roig *et al*, 2010; Moehle *et al*, 2012; Gaytán *et al*, 2013). Importantly, our genetic analysis places Euc1 in a functional pathway with the Rpd3S complex (Fig 8D and E). Rpd3, an orthologue of mammalian HDAC1 family enzymes, which generally repress transcription, was shown to be recruited to promoters of osmo- and heat stress-responsive genes, mostly as part of the Rpd3L complex, to activate their expression under stress conditions (de Nadal *et al*, 2004; Yang & Seto, 2008; Ruiz-Roig *et al*, 2010). On the other hand, the Rpd3S complex mainly acts within transcribed regions and is recruited by Set2-mediated H3K36 methylation to deacetylate histone H4 and establish a repressed state after transcription to prevent spurious cryptic transcription (Carrozza *et al*, 2005). More recently, cryptic transcripts regulated by this pathway have been demonstrated to also regulate promoter states and transcription of coding transcripts, in particular upon changing nutrient or stress conditions and in aged cells (Sen *et al*, 2015; Kim *et al*, 2016; McDaniel *et al*, 2017).

While the precise mechanism by which Euc1 and Rpd3S cooperate is currently elusive, our data provide clues to guide future research: In *euc1∆* cells, several genes are misregulated, including *RCO1* of the Rpd3S complex (upregulated) and *SIR2*, a sirtuin family HDAC (downregulated). Strikingly, *HSP12* was downregulated in *euc1∆* and upregulated upon *EUC1* overexpression. Hsp12 is involved in maintaining membrane organization and is a direct target of Rpd3-dependent activation upon osmotic stress (de Nadal *et al*, 2004). Deregulation of cellular membrane homeostasis could also provide a rationale for the observed phenotypes of *EUC1* overexpression strains upon exposure to benzyl alcohol, which has been described to interfere with membrane organization (Lone *et al*, 2015). Additionally, DMSO, which enhances some genetic interactions, can interfere with membrane organization and cells require the H2.Z pathway for DMSO tolerance (Gurtovenko & Anwar, 2007; Gaytán *et al*, 2013). Of note, DMSO was also found to cause weak HDAC inhibition (reviewed in Marks & Breslow, 2007), and might thereby also aggravate gene expression defects in already partially compromised backgrounds. In all, our data suggest a model whereby several gene regulatory mechanisms, including ub-hotspots and Rpd3S, have overlapping functions to allow adaptation to different conditions of cellular stress.

## Specificity in the STUbL pathway is achieved by multivalent substrate–ligase contacts

Slx5-SIMs are required for ubiquitylation of all currently known Slx5/Slx8 substrates. In yeast, several phenotypes such as hydroxyurea sensitivity, as well as accumulation of high-molecular-weight SUMO conjugates, can be complemented by the distantly related mammalian RNF4 or even Arkadia/RNF111 variants (Prudden *et al*, 2007; Sun *et al*, 2007, 2014; Uzunova *et al*, 2007). Because complementation is strictly SIM-dependent, these functions may rely on a "polySUMO-SIM interaction mode", which would exclusively involve recognition of polySUMO chains by multiple SIMs, but not necessarily additional substrate recognition features within the different STUbLs. Also in the case of Matα2, intact SIMs of Slx5 have been shown to be required for Matα2 turnover, even though SUMOylation is not required for STUbL-dependent ubiquitylation of Matα2, possibly because Slx5-SIMs recognize hydrophobic features on Matα2 (Xie *et al*, 2010).

In case of Euc1, we have not observed any evidence for long SUMO chains attached to Euc1, suggesting that a "polySUMO-SIM interaction mode" is unlikely to account for Euc1 recognition. In contrast, our data suggest that not only the SUMO-SIM interaction, but also additional specific contacts between the central part of Euc1 and the Slx5-Md are needed for ubiquitylation of Euc1 and possibly other substrates at the ub-hotspots. It seems intuitive that these interaction surfaces will collectively allow a "bipartite recognition mode" and provide the required affinity/avidity for specific Slx5/Slx8 recruitment. Bipartite substrate recognition has so far not been demonstrated for Slx5/Slx8. However, it is a well-known mode of interaction in SUMO-regulated pathways, a prominent example being recognition of PCNA-SUMO by the helicase Srs2 (Papouli *et al*, 2005; Pfander *et al*, 2005; Armstrong *et al*, 2012).

Notably, bipartite substrate recognition is a more general theme for STUbL enzymes: RNF4 has a basic patch that mediates targeting to nucleosomes (Groocock *et al*, 2014), as well as an arginine-rich motif required for interaction with phosphorylated substrates (Kuo *et al*, 2014; Thomas *et al*, 2016). Arkadia/RNF111 uses its Mn/Mc domains for substrate interaction and localization (Sun *et al*, 2014). Degringolade/Dgrn uses its RING domain for interaction with Hairy (Abed *et al*, 2011). Euc1 recognition by Slx5/Slx8 may therefore serve as a guide for future research elucidating how different substrate recognition domains control the diverse STUbL functions from the response to DNA damage to early embryonic development.

# Materials and Methods

## Yeast and molecular biology methods

All yeast and molecular biology methods followed standard procedures and are specified in the Appendix Supplementary Methods.

## ChIP and ChIP-on-chip (ChIP-chip) analysis

ChIP experiments were performed as described previously with minor modifications (Aparicio *et al*, 2005; Kalocsay *et al*, 2009). Briefly, cells were grown to mid-log phase, crosslinked with formaldehyde, and chromatin was isolated and enriched for the indicated proteins with specific antibodies. Subsequently, input and enriched DNA were quantified by qPCR on a Light Cycler 480 System (Roche) or analyzed on yeast tiling arrays (NimbleGen) for genome-wide binding data. Genome-wide binding profiles (ChIP-chip) were generated from two independent experiments including a dye-swap, except for ubiquitin (FK2) *rad6∆* and IgG WT profiles in Fig 1A. All ChIP-qPCR data presented are means ± SD from 2 to 5 independent experiments with $> 10^9$ cells and triplicate qPCR quantification. Where *P*-values are given, an unpaired, two-tailed

Student's *t*-test was applied. See Appendix Supplementary Methods for details.

### Yeast one-hybrid (Y1H) screen and reporter gene assays

To isolate proteins binding to the ub-HS-motif, two independent Y1H screens were performed. For screen 1, the reporter strain carrying a *ubx5Δ* allele (MJK391) was derived from YM4271 (Clontech). For screening, three copies of the mapped ub-HS4-motif (chromosome XIII 308901-308939) were cloned upstream of a minimal promoter followed by *HIS3* (cloned from pHISi-1 (Clontech)) and integrated at the *URA3* locus. A yeast cDNA library cloned into a vector with an N-terminally fused Gal4 activation domain (AD) (Dualsystems) was transformed into the strain, and clones were selected on media lacking histidine supplemented with 50 mM 3-amino-triazole (3AT). Positive clones were isolated, retested, and sequenced. Screen 2 was performed by Hybrigenics Services SAS (Paris, ULTImate Y1H Screen), with three copies of chromosome XIII 308898-308939 upstream of a *HIS3* reporter gene in *UBX5* cells and 2 mM 3AT selection.

For Y1H/reporter gene assays, either plasmids encoding AD-fusion constructs or untagged proteins were transformed, several clones were mixed, adjusted to $OD_{600} = 0.5$, and fivefold serial dilutions were spotted on control or selective media and incubated at 30°C for 2–5 days as indicated.

### Transcriptome analysis (RNAseq)

Total RNA was isolated as detailed in Appendix Supplementary Methods, polyA RNA was enriched (NEBNext Poly(A) mRNA Magnetic Isolation Module #E7490), and libraries were prepared for sequencing (NEBNext Ultra II RNA library prep Set for Illumina #E7770L) and barcoded (NEBNext Multiplex Oligo for Illumina #E7335L, #E7500L, #E7710L, #E7730L), all according to the manufacturer's instructions. 75-bp single-end reads were obtained by sequencing on an Illumina NextSeq 500 instrument using a NextSeq 500/550 High Output Kit v2 (75 cycles, Illumina). Sequencing reads were aligned to the yeast transcriptome (ENSEMBL R64-1-1, annotation version 94) using STAR (v. 2.6.0a). Read counts per gene were provided by STAR, and TPM expression values were calculated with RSEM (v. 1.3.0). We used the unfiltered count table for differential expression analysis in DESeq2 (v. 1.22.2). Based on the standard pipeline, we estimated size factors and dispersion values for each gene and fitted a generalized linear model with a single factor "genotype". Significance testing was based on the Wald test (default parameters). The results were extracted with an alpha of 0.1, an lfcThreshold of 0, and independent filtering (default parameters).

Microarray-based transcriptome analysis was performed using GeneChIP Yeast Genome 2.0 arrays (Affymetrix) and is detailed in the Appendix Supplementary Methods.

### Biochemical methods

For co-immunoprecipitation, cells were lysed under native buffer conditions and cleared lysate was subjected to immunoprecipitation with either specific Euc1 antibody or anti-FLAG resin (M2, Sigma). Eluted proteins were probed by Western blot (WB). See Appendix Supplementary Methods for details.

To detect proteins covalently modified by either SUMO or ubiquitin, yeast strains expressing either N-terminally histidine-tagged SUMO (7 histidines) or ubiquitin (6 histidines) under the control of the *ADH1* promoter, integrated at the *URA3* locus, were used. Subsequently, NiNTA-PDs using either NiNTA agarose (Figs 4E and EV2B) or magnetic agarose beads (all others, both Qiagen) were performed as described (Psakhye & Jentsch, 2016).

Additional materials and methods can be found in the Appendix Supplementary Methods; for yeast strains, see Appendix Table S1, plasmids Appendix Table S2, qPCR primers Appendix Table S3.

## Data availability

Yeast strains and plasmids are available on request. ChIP-chip, RNAseq, and microarray data are available from Gene Expression Omnibus (GEO, https://www.ncbi.nlm.nih.gov/geo/) entry GSE118818.

**Expanded View** for this article is available online.

## Acknowledgements

We thank S. Brill, A. Buchberger, and M. Hochstrasser for materials; Sven Schkölziger, Alexander Strasser, Katrin Strasser, Jochen Rech, Florian Paasch, and Ludwig Rieger for technical assistance; Kerstin Maier and Assa Yeroslaviz for help with microarray analysis; Marja Driessen for help with RNA sequencing; Julian Stingele and Florian Wilfling for providing data; Johannes Heipke for plasmid construction; Julian Stingele, Ivan Psakhye, Neysan Donnelly, Florian Paasch, and members of the Jentsch and Pfander laboratories for fruitful discussions and critical reading of the manuscript.

Research in the B.P. laboratory is supported by Max Planck Society, the collaborative research cluster (SFB1064), and project grants from the German Research Council (DFG). Research in the S.J. laboratory is supported by Max Planck Society, Deutsche Forschungsgemeinschaft, Center for Integrated Protein Science Munich, Louis-Jeantet Foundation, and a European Research Council (ERC) Advanced Grant (#339176). M.J.K. was supported by a fellowship of the Boehringer Ingelheim Fonds (BIF).

## Author contributions

MH, MJK, BP, and SJ designed experiments and analyzed data. MH performed experiments for Figs 1B and D–F, and 2E, G and H, and 3 and 4C, E and F, and 4–7 and EV1H and EV2A–G and I, and EV3B and C, and EV4, Appendix Figs S2E, S3–S7, S8A–C and E, and S9–S12. MJK performed experiments for Figs 1A and B, and F–G and 2A–D and F, and 4A and B, and D, and EV1A–G and EV2H and EV3A and B, Appendix Figs S1, S2A–D and S8D. TS performed analysis of ChIP-chip and RNAseq data. RP and BHH performed analysis of *de novo* motif searches. MH and BP wrote the manuscript, and all authors commented on the manuscript.

## Conflict of interest

The authors declare that they have no conflict of interest.

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
