## [Review Process File · The EMBO Journal]

Slx5/Slx8-dependent ubiquitin hotspots on chromatin contribute to stress tolerance

Markus Höpfler, Maximilian J. Kern, Tobias Straub, Roman Prytuliak, Bianca H. Habermann, Boris Pfander and Stefan Jentsch.

Review timeline:

Submission date:	27 th July 2018
Editorial Decision:	31 st August 2018
Revision received:	12 th February 2019
Editorial Decision:	15 th March 2019
Revision received:	29 th March 2019
Accepted:	3 rd April 2019

Editor: Hartmut Vodermaier

Transaction Report:

1st Editorial Decision

31st August 2018

Thank you again for submitting your manuscript on Slx5/8 and Euc1-linked ubiquitin hotspots on chromatin for our editorial consideration. I have now obtained reviews from three trusted experts in the field, copied below for your information. As you will see, the referees acknowledge the technical quality of the study, as well as the potential interest of your findings. However, I am afraid that all three also share the overriding key criticism that the functional significance of Euc1 and the chromatin ubiquitin hotspots remains at this stage still very unclear. While I understand that this seems to have been difficult to address, I am afraid that this shared major concern leaves us little choice but to conclude that we cannot offer to publish the paper in its present form in The EMBO Journal. I realize that it may well become a more compelling candidate if you should be able to extend it and obtain some better functional understanding, and at least referees 2 and 3 offer a number of constructive suggestions for starting points of such further investigations; but given that it is unclear whether these will indeed lead to sufficiently conclusive insights in a timely manner, I do not feel that this would fall within the scope of a regular revision, and therefore would not allow us to provide the commitment we usually make by inviting revision. That said, should the guidance provided by our reviewers indeed allow you to obtain more definitive information on the roles of Euc1 and/or ubiquitin hotspots, I'd be more than happy to look at a new version of the study once again!

I am sorry that the reviews did not allow me to be more positive for The EMBO Journal at this stage, but would nevertheless like to thank you again for the opportunity to consider this work. I hope that you may find our referees' comments helpful for possibly extending the study, or that you may alternatively consider the opportunity to publish in Life Science Alliance worthwhile.

REFERE REPORTS

Referee #1:

Höpfler et al show that yeast chromatin contains a small number of ubiquitin "hotspots" that represent defined DNA loci. The SUMO Targeted Ubiquitin Ligase (STUbL) Slx5/Slx8 is recruited to the same DNA loci and is responsible for the accumulation of ubiquitin at these loci. Recruitment of Slx5/Slx8 is dependent on the transcription factor Euc1. What is demonstrated is that the STUbL recruitment is not just dependent on SUMO modification of Euc1, but also involves protein-protein interactions between Slx5 and Euc1. This bipartite recruitment results in the deposition of ubiquitin at the specific DNA loci that contain binding sites for the Euc1 transcription factor. The work that is presented is nicely done and the experiments are all fairly clear cut. The problem with the paper is that neither SUMO modification of the transcription factor nor the deposition of ubiquitin appear to have any biological consequences. It also seemed that the paper lacked any biochemical evidence for direct modification by SUMO/ubiquitin and direct interactions. For instance using recombinant proteins the DNA binding activity on specific and non-specific DNA sequences of the free and the SUMO modified Euc1 transcription factor could have been accurately evaluated. The ability of Slx5/8 to ubiquitinate such DNA bound proteins could also have been determined. Without some functional data it is difficult to recommend publication of this paper in EMBO J.

Referee #2:

The manuscript by Höpfler et al. reports interesting and unexpected findings on distinct regions of yeast chromatin that are enriched in ubiquitin, which they call "ubiquitin hotspots" (ub-hotspots). These ub-hotspots depend on the SUMO-targeted ubiquitin ligase (STUbL) enzyme Slx5/8 and a SUMO-modified form of a protein named Euc1, which is characterized for the first time in this manuscript. Protein dynamics at ub-hotspots are also affected by the Cdc48 machinery, which has previously been connected to Slx5/8 as well as ubiquitin- and SUMO-dependent chromatin dynamics.

While the data in the manuscript are solid, interesting, and nicely presented, questions remain about the significance of the main findings. The Euc1 protein is vital for ub-hotspot formation, yet a functional role for ub-hotspots or Euc1 in yeast is lacking—other than a proposed vague role for Euc1 in transcription. The manuscript provides no evidence that loss of Euc1 function results in a clear cellular phenotype, whereas cells lacking Slx5/8 activity have many pronounced phenotypic defects. The title of the manuscript indicates that the ub-hotspots have a role in STUbL specificity, but the data suggest that ub-hotspots are not important for Slx5/8 function other than that related to Euc1 (though not all Slx5/8 functions were tested in the manuscript). So any current argument for ub-hotspot function becomes 'circular'. Therefore, I suggest the authors focus on providing some cellular function or physiological need for Euc1 and/or ub-hotspots. My specific comments and suggestions are below.

Major comments:

1) Since the main weakness of the manuscript is a lack of functional data for ub-hotspots or the Euc1 protein, I suggest a focus on generating data related to yeast physiology. For example, earlier genomic-scale genetic interaction datasets suggest that lack of the EUC1 gene apparently worsens poor growth of strains with mutations in genes of known function (see below). In theory, one could also mutate all 7 ubiquitin hotspots (with CRISPR, for example), although this is beyond what would be expected for publication of the current study.

I performed searches for "YMR111C" (Euc1) using the SGD and BioGRID databases with some interesting results. First, a physical interaction between Slx5 and Euc1 was previously reported (Yu et al., 2008). More generally, several "high-throughput" studies have connected Euc1 to transcription, consistent with some of the data presented in Höpfler et al. (SGD and BioGRID). There are also negative genetic interactions between euc1 and mutations in components of the ubiquitin-proteasome system (UPS): DOA1, SEM1, SAN1, RPN12, and RPT6 (Zheng et al., 2010; Costanzo et al., 2016).

It would be nice if the authors could confirm and extend some of these results. Since specific data

from high-throughput studies are often difficult to mine and visualize, I would suggest performing dilution series growth assays for some of the above mentioned genetic interactions. A robust growth disadvantage of a strain lacking EUC1 would also allow testing functionality of various *eucl* alleles (such as the SUMO-deficient allele or the Slx5-interaction mutants, which would connect the data back to Slx5/8). More importantly, these studies could lead to insights into Euc1 function. Linking specific transcription machinery to Euc1 may hint at its more specific function in transcription. With regards to the UPS, were the particular UPS components connected to EUC1 linked because of their unique presence (or enrichment) in the high-throughput libraries used or are they uniquely related to Euc1? Perhaps most interesting are UPS components providing the most specificity, namely the E3 San1—which is known to function in nuclear protein quality control, like Slx5/8. Even data showing genetic interactions of EUC1 with more general components of the UPS (like Doa1 or the proteasome) would be nice.

2) A previous study reported ChIP-chip results using the Slx5 subunit of Slx5/8, rather than the Slx8 subunit as in the current manuscript (van de Pasch et al., 2013). In the published study, Slx5 localized to each centromere of the 16 yeast chromosomes. Interestingly, Slx8 did not show this localization. Some discussion of these results should be included in the current manuscript. Also, the 2013 paper (along with a few other papers on Slx5/8), provided phenotypes of *slx5Δ* and *slx8Δ* strains that could be investigated in relation to ub-hotspots and/or Euc1.

3) On page 14, it is speculated that Euc1 might change transcription based on some stimulus. Perhaps a survey of cell growth conditions (different temperatures, different chemicals) using the *euclΔ* strain, or a double mutant including *euclΔ* (see point number one above), might hint at this putative stimulus.

4) The authors should address whether there may be ub-hotspots and/or Euc1 equivalents in other organisms/STuBL systems. It was not mentioned whether the ub-HS-motif was found in other genomes (besides *Saccharomyces cerevisiae*). There was also no mention of the evolutionary distribution of Euc1. The authors did mention human STuBLs, including RNF4. I believe that RNF4 does not have a sequence like that of the Slx5-Md. This would suggest that RNF4, which can complement loss of Slx5/8 in yeast based on growth phenotypes, would not function in yeast ub-hotspot formation (or Euc1 binding). This could be tested.

Minor comments:

1) I'm not so fond of the part of the discussion about 'SUMO-SIM interaction only' mode. Data showing that a STuBL-sim mutant does not complement a certain phenotype does not rule out other substrate-recognition features involved in a particular interaction. In the manuscript, both the Slx5-SIM* and Slx5-MdΔ mutants appear to be fully defective for ub-hotspot formation (Figure 5). It may be that most STuBL functions involve additional interactions other than SUMO-SIM interactions, as alluded to in the final paragraph of the discussion. Simply based on protein size, I would think that Slx5/8 and Arkadia have more macromolecular interactions than RNF4. I would change the wording at the end of the discussion a bit, maybe removing the "only" and focusing on the SIM-independent interactions by STuBLs, with Euc1 being another example.

REFERENCES

- Costanzo et al. (2010) *Science*. 327 (5964): 425-31.
- Costanzo et al. (2016) *Science*. 353 (6306). pii:aaf1420
- van de Pasch et al. (2013) *PLoS One*. 8 (6): e65628
- Yu et al. (2008) *Science*. 322 (5898):104-10.
- Zheng et al. (2010) *Mol Syst Biol*. 6: 420

Referee #3:

To investigate Slx5/Slx8 STUbL-mediated ubiquitylation of chromatin-associated proteins in budding yeast, the authors carried out ChIP-seq for Ub itself and Slx8. Ub signals were found widely across the genome, with signals being increased in a *cdc48-6* mutant background, presumably as a result of increased H2B-Ub levels. In contrast, Slx8 was only localized to a few sites, overlapping with strong Ub signals at seven discrete intergenic sites on four different chromosomes. They showed that K48 Ub chains were present at these sites, and that this was dependent on functional Slx5 and Slx8. Analysis of the seven sites revealed a conserved 36 bp motif, and using a 1-hybrid screen to identify motif-binding proteins they identified YMR11, which they renamed EUC1 (Enriches Ubiquitin on Chromatin), as well as SLX5 and SMT3/SUMO. Using anti-Euc1 antibodies for ChIP-seq they showed that Euc1 was primarily located at these seven hotspots, and that mutation of the consensus motif abolished Euc1 binding. Euc1 possesses a Gcr1 domain at the C-terminus, which is known to have sequence-specific DNA-binding activity, and they showed that Euc1 could drive expression of a HIS3 reporter gene equipped with the binding motif upstream of HIS3 gene; reduced reporter activity was noted in a *euc1Δ* strain. However, expression of the genes surrounding the chromosomal hotspots was unaffected in the *euc1Δ* strain. They mapped a transactivation domain in Euc1 to an acidic region containing residues 19-28. They showed that Euc1 was sumoylated predominantly at K231, lying in a SUMO consensus motif, generating mono-sumoylated Euc1 species, and that Ub-hotspot formation did not occur in a K231R Euc1 strain. A *Siz1/Siz2* double knockout strain lacked Ub hotspots, suggesting that *Siz1/Siz2* are the major Euc1 SUMO E3 ligases. Euc1 was shown to be polyubiquitylated in a Slx5/Slx8-dependent manner, consistent with the ubiquitylation being carried out by Slx5/Slx8-mediated STUbL activity, and the demonstration that Euc1 was required for Slx5/Slx8 recruitment to the hotspots. Slx5/Slx8 recruitment to substrates is often dependent on interaction of its tandem SIMs with SUMO chains, but Euc1 is only mono-sumoylated and they showed that in addition to the Slx5 SIM interaction with SUMO-K231 there is also a direct interaction of Euc1 and Slx5 between a region containing a coiled-coil dimerization motif and downstream sequences in the middle of Euc1, and the M-domain lying between the SIMs and the RING domain in Slx5. Mutations in Euc1 that abolished Slx5 binding reduced ubiquitylation of Euc1 itself and also the formation of the Ub hotspots. Overall, they conclude that a multivalent interaction between Euc1 and Slx5 mediates Slx5/Slx8 recognition of Euc1 as a SUMO substrate for ubiquitylation.

This is a very nice paper showing that the Euc1 transcription factor binds to a minimal set of discrete sites at seven chromosomal locations in the *S. cerevisiae* genome. Euc1 bound at these sites is in turn sumoylated by the *Siz1/Siz2* SUMO ligases, resulting in the recruitment of the Slx5/Slx8 STUbL, leading to local polyubiquitylation of Euc1 and other chromatin components. The data are convincing, but what is missing is the true functional significance of this series of events at these seven chromosomal Ub hotspots, i.e. what is the physiological function of these Ub hotspots. No strong phenotype was noted with the *euc1Δ* strain, or a significant difference in the transcription of genes adjacent to the hotspots.

Points: 1. To what extent are Euc1 itself and the hotspots conserved in other yeasts or in higher eukaryotes?

2. What are the phenotypes of *euc1Δ* cells, for instance when they are subjected to stress?

3. Page 7, line 223: Here, the text and the data in Figure S3G do not seem consistent with each other. Deletion of *ubx5* conferred a mild activation of the HIS3 reporter gene expression (although a WT cell control should be shown), which was abolished by *euc1* deletion. Here, an experiment using the same ub-HS4-HIS3 reporter assay, comparing WT with *euc1-KR*, *siz1Δ siz2Δ*, *slx5Δ*, *slx8Δ*, *cdc48-3*, *euc1Δ*, as well as *euc1Δ* combined with *siz1*, *siz2*, *slx5*, *slx8*, and *cdc48* mutants would be informative. If the model is correct, HIS3 expression may be turned on when the pathway is disrupted, and then turned back off when *euc1* is deleted. This result would be a good addition to support their proposed model regarding its physiological function.

4. Does n-terminal fusion of SUMO to K231R-Euc1 rescue the hotspot defect?

5. Is anything known about what de-sumoylates SUMO-Euc1 - Ulp1/2?

6. Could Euc1 drive transcription of a non-coding RNA?

We would like to thank all reviewers for their time and constructive criticism towards our paper. We understand that all three reviewers found the data presented in the first version of our paper clear and the experiments well done, but that they noted that the initial version of our paper did not contain any data on the function of Euc1 or the ubiquitin hotspots (ub-hotspots). In the revised version, we have therefore concentrated our efforts on addressing this principal criticism and have added two additional figures that clearly show a function of Euc1, and importantly the Euc1-dependent ub-hotspot mechanism, in thermotolerance and resistance against other stresses, which is linked to transcriptional regulation by the Rpd3S complex.

Reviewer #1:

Hopfler et al show that yeast chromatin contains a small number of ubiquitin "hotspots" that represent defined DNA loci. The SUMO Targeted Ubiquitin Ligase (STUbL) Slx5/Slx8 is recruited to the same DNA loci and is responsible for the accumulation of ubiquitin at these loci. Recruitment of Slx5/Slx8 is dependent on the transcription factor Euc1. What is demonstrated is that the STUbL recruitment is not just dependent on SUMO modification of Euc1, but also involves protein-protein interactions between Slx5 and Euc1. This bipartite recruitment results in the deposition of ubiquitin at the specific DNA loci that contain binding sites for the Euc1 transcription factor. The work that is presented is nicely done and the experiments are all fairly clear cut. The problem with the paper is that neither SUMO modification of the transcription factor nor the deposition of ubiquitin appear to have any biological consequences. It also seemed that the paper lacked any biochemical evidence for direct modification by SUMO/ubiquitin and direct interactions. For instance using recombinant proteins the DNA binding activity on specific and non-specific DNA sequences of the free and the SUMO modified Euc1 transcription factor could have been accurately evaluated. The ability of Slx5/8 to ubiquitinate such DNA bound proteins could also have been determined.

Without some functional data it is difficult to recommend publication of this paper in EMBO J.

We agree with the reviewer that our initial manuscript did not include any functional data, but we do not think that it suggested that the ub-hotspot mechanism would not have any biological consequence. We have now filled this important gap by adding two additional figures on Euc1 function (Fig. 6-7), which show that Euc1 is required for growth under conditions of heat/cold stress or DMSO stress, if cells additionally lack factors that reduce transcriptional fidelity. We show that Euc1 can be assigned to a pathway involving the histone deacetylase (HDAC) complex Rpd3S, which is involved in repressing spurious transcription within gene bodies and has been linked to cellular stress responses before. Importantly, by complementing *euc1Δ* phenotypes with our previously characterized mutant versions, we can show that it is specifically the ub-hotspot mechanism involving Euc1-SUMOylation and Slx5/Slx8 that is critical for this function and not Euc1's putative transcription factor function.

With these additional data our manuscript has become quite extensive, and while we appreciate the additional information that could be gathered from the suggested biochemical characterization, we therefore reckoned that an *in vitro* reconstitution of ub-hotspots would be beyond the scope of this manuscript.

Reviewer #2:

The manuscript by Höpfler et al. reports interesting and unexpected findings on distinct regions of yeast chromatin that are enriched in ubiquitin, which they call "ubiquitin hotspots" (ub-hotspots). These ub-hotspots depend on the SUMO-targeted ubiquitin ligase (STUbL) enzyme Slx5/8 and a SUMO-modified form of a protein named Euc1, which is characterized for the first time in this manuscript. Protein dynamics at ub-hotspots are also affected by the Cdc48 machinery, which has previously been connected to Slx5/8 as well as ubiquitin- and SUMO-dependent chromatin dynamics.

While the data in the manuscript are solid, interesting, and nicely presented, questions remain about the significance of the main findings. The Euc1 protein is vital for ub-hotspot formation, yet a functional role for ub-hotspots or Euc1 in yeast is lacking^{3/4}other than a proposed vague role for Euc1 in transcription. The manuscript provides no evidence that loss of Euc1 function results in a clear cellular phenotype, whereas cells lacking Slx5/8 activity have many pronounced phenotypic defects. The title of the manuscript indicates that the ub-hotspots have a role in STUbL specificity, but the data suggest that ub-hotspots are not important for Slx5/8 function other than that related to Euc1 (though not all Slx5/8 functions were tested in the manuscript). So any current argument for ub-hotspot function becomes 'circular'. Therefore, I suggest the authors focus on providing some cellular function or physiological need for Euc1 and/or ub-hotspots. My specific comments and suggestions are below.

Major comments:

1) Since the main weakness of the manuscript is a lack of functional data for ub-hotspots or the Euc1 protein, I suggest a focus on generating data related to yeast physiology. For example, earlier genomic-scale genetic interaction datasets suggest that lack of the EUC1 gene apparently worsens poor growth of strains with mutations in genes of known function (see below). In theory, one could also mutate all 7 ubiquitin hotspots (with CRISPR, for example), although this is beyond what would be expected for publication of the current study.

I performed searches for "YMR111C" (Euc1) using the SGD and BioGRID databases with some interesting results. First, a physical interaction between Slx5 and Euc1 was previously reported (Yu et al., 2008). More generally, several "high-throughput" studies have connected Euc1 to transcription, consistent with some of the data presented in Höpfler et al. (SGD and BioGRID). There are also negative genetic interactions between euc1 and mutations in components of the ubiquitin-proteasome system (UPS): DOA1, SEM1, SAN1, RPN12, and RPT6 (Zheng et al., 2010; Costanzo et al., 2016).

It would be nice if the authors could confirm and extend some of these results. Since specific data from high-throughput studies are often difficult to mine and visualize, I would suggest performing dilution series growth assays for some of the above mentioned genetic interactions. A robust growth disadvantage of a strain lacking EUC1 would also allow testing functionality of various euc1 alleles (such as the SUMO-deficient allele or the Slx5-interaction mutants, which would connect the data back to Slx5/8). More importantly, these studies could lead to insights into Euc1 function. Linking specific transcription machinery to Euc1 may hint at its more

specific function in transcription. With regards to the UPS, were the particular UPS components connected to EUC1 linked because of their unique presence (or enrichment) in the high-throughput libraries used or are they uniquely related to Euc1? Perhaps most interesting are UPS components providing the most specificity, namely the E3 San1 which is known to function in nuclear protein quality control, like Slx5/8. Even data showing genetic interactions of EUC1 with more general components of the UPS (like Doa1 or the proteasome) would be nice.

We highly appreciate the reviewer's comments, his efforts and the very valuable suggestions. Indeed, we manually tested a large set of genetic interactions from high-throughput studies. While we could not reproduce all synthetic phenotypes from previous large-scale studies (possibly due to effects on the neighbouring gene *MED11* in previous studies, see Fig. S11C-F), we were able to confirm negative genetic interactions of *EUC1* mostly with several genes involved in chromatin organization or gene expression regulation, as suggested by gene ontology term enrichments of genetic interactors (Supplementary table S9). We observed interactions with the H2A.Z-SWR1-C pathway, the RNA processing and transport factor *NPL3*, and the oxidative stress-responsive transcription factor (TF) *STB5*, most of which appeared specifically upon exposure of cells to stress conditions including thermostress (cold or heat) or exposure to chemicals like DMSO (Fig. 7A, S11A-F). The strongest genetic interactions manifested with members of the Rpd3L HDAC complex upon exposure to heat or DMSO stress (*DEP1*, *RXT2* and *SDS3*, Fig. 7B-C). The genetic interaction with the Rpd3L HDAC was very informative: the Rpd3L and the Rpd3S complex show strong negative genetic interactions as well, most likely due to functional overlap and using epistasis analysis we were able to assign Euc1 to a pathway involving the Rpd3S HDAC (Fig. 7D-E).

We note that we could not confirm any growth phenotypes of *euc1Δ* when paired with other interactors from previous high throughput studies, including most ubiquitin proteasome system components (*DOA1*, *SEMI*, *SAN1*, *RPT6* (*cim3-1*), weak negative interaction with *rpn12-1*, Fig. S11A).

Critically, we followed the reviewer's suggestion to test our specific *euc1*-alleles for their ability to complement these genetic interactions. Specifically, we used the *sds3Δ* (Rpd3L) and *npl3Δ* backgrounds and found that variants of *EUC1* that could not form ub-hotspots did not complement the synthetic growth phenotypes (Fig. 7G, S13A). In contrast, variants of *EUC1* that lacked an N-terminal transactivation domain rescued any growth defect (*N30Δ*, Fig. 7G, S13A). These experiments therefore allowed us to conclude that the function of Euc1 at ub-hotspots is critical for *euc1* phenotypes and likely the function in the Rpd3S pathway, while the ability of Euc1 to act as transcriptional activator is not.

Lastly, we also tested overexpression of Euc1 and found a pronounced toxicity especially at 37°C, upon exposure to the membrane-fluidizing drug benzylalcohol or in absence of Slx5/Slx8 (Fig. 6C, S10B). This toxicity appears to involve DNA-binding and transactivation at sites other than ub-hotspots (Fig. 6D-E, S10F-G). Interestingly, simultaneous overexpression of *SLX5/SLX8* could mitigate the phenotypes suggesting that the STUbL can repress spurious Euc1 activation (Fig. 6F, S10H). Overall, our new data lead us to propose that Euc1 function needs to be tightly controlled and that the critical function of Euc1 is linked to ub-hotspots rather than a direct transactivation of neighboring genes by Euc1.

2) A previous study reported ChIP-chip results using the Slx5 subunit of Slx5/8, rather than the Slx8 subunit as in the current manuscript (van de Pasch et al., 2013). In the published study, Slx5 localized to each centromere of the 16 yeast chromosomes. Interestingly, Slx8 did not show this localization. Some discussion of these results should be included in the current manuscript. Also, the 2013 paper (along with a few other papers on Slx5/8), provided phenotypes of *slx5D* and *slx8D* strains that could be investigated in relation to ub-hotspots and/or Euc1.

We examined our ChIP-chip data for Slx8, but as described in van de Pasch et al. (2013) for Slx8, we did not find any centromeric enrichments. We also tested the localization of both Slx5 and Slx8 to centromeres *CEN1* and *CEN11* in ChIP-qPCR experiments, but again could not observe any specific enrichment, neither with tagged alleles nor our specific Slx5 and Slx8 antibodies in WT strains (Fig. R1A-C). It is possible that the observed discrepancies might be due to the Slx5-GFP construct used in the previous study (van de Pasch et al. (2013), as also the authors state in their methods section (which we now comment in the legend to Fig. S2G):

“All epitope- or fluorescent-tagged strains exhibited wt growth with the exception of Slx5-GFP. All attempts to fuse SLX5 to a variety of tags, either C- or N-terminally, resulted in strains with slow growth. All other epitope-tagged strains exhibited wt growth.”

In this regard, we would like to point out that we have also observed growth phenotypes when attempting C-terminal tagging of Slx5, but that the strains used in our experiments (expressing 3HA-Slx5 or Slx8-9myc, respectively) did not exhibit any growth or hydroxyurea phenotype in our hands (Fig. R1D). As Slx5/Slx8 has substrates with centromeric or kinetochore-localization (Ohkuni *et al*, 2016; Schweiggert *et al*, 2016) it could be speculated that a non-functional Slx5 (such as Slx5-GFP) might become trapped at centromeres.

Figure R1. ChIP-qPCR experiments of 3HA-Slx5 (A), Slx8-9myc (B) with tag-specific antibodies or endogenous Slx5 and Slx8 using polyclonal antibodies (C). (D) Growth assay of the indicated strains, including strains used in (A-B).

Furthermore, while at this point we cannot exclude a role of ub-hotspots in genome stability, we did not follow up this lead, as *slx5-Md* Δ fully complements the *slx5* Δ HU sensitivity, but does not restore ub-hotspots (Fig. 5E, S8A). In contrast, the ub-hotspot function appears to contribute to *SLX5* phenotypes as we observed a highly similar negative genetic interaction between *SLX5* and *SDS3* as for *EUC1* and *SDS3*, and complementation depends on the Slx5-Md (Fig. 7H, S13C-D).

3) On page 14, it is speculated that Euc1 might change transcription based on some stimulus. Perhaps a survey of cell growth conditions (different temperatures, different chemicals) using the *euc1D* strain, or a double mutant including *euc1D* (see point number one above), might hint at this putative stimulus.

Indeed, we have now discovered several conditions, when Euc1 function becomes critical for cellular survival (Fig. 7). However, our complementation analysis suggests that Euc1's putative transcription activator function is still not critical under these conditions (Fig. 7G, S13A). As additionally, we do not find evidence for direct transcriptional activation of ub-hotspot adjacent genes, we now propose that Euc1's role at ub-hotspots is independent of a putative role as transcription activator.

4) The authors should address whether there may be ub-hotspots and/or Euc1 equivalents in other organisms/STuBL systems. It was not mentioned whether the ub-HS-motif was found in other genomes (besides *Saccharomyces cerevisiae*). There was also no mention of the evolutionary distribution of Euc1. The authors did mention human STuBLs, including RNF4. I believe that RNF4 does not have a sequence like that of the Slx5-Md. This would suggest that RNF4, which can complement loss of Slx5/8 in yeast based on growth phenotypes, would not function in yeast ub-hotspot formation (or Euc1 binding). This could be tested.

We apologize that our initial manuscript failed to comment on the aspect of evolutionary conservation, we now address this in Fig. 2I-J and Fig. S3G-H. Briefly, clear Euc1-homologs can be easily detected in several other yeast species and conservation is most striking in the DNA-binding domain and around the coiled-coil domain. In related *Saccharomyces* species we can also observe the ub-hotspot motif and we found at least one ub-HS-motif for each ub-hotspot, with particular conservation of the consensus motif. In more distantly related species it becomes difficult to prove the presence of ub-HS motifs as genome structure is diverging. We therefore cannot comment on conservation in higher eukaryotes.

We tested if RNF4 or Arkadia could complement a lack of *SLX5/SLX8* when expressed in yeast. While both RNF4 and a variant of Arkadia/RNF111 complement the *slx5* Δ /*slx8* Δ HU-sensitivity (Uzunova *et al*, 2007; Sun *et al*, 2007; 2014; Prudden *et al*, 2007), ub-hotspots were not restored with these constructs (Fig. R2A-B). Furthermore, in an effort to 'transplant' the Slx5-Md to RNF4 or Arkadia, we created several RNF4-Slx5-Md hybrid constructs. These constructs could however not restore ub-hotspot function, even though most of them complemented *slx5* Δ HU- and cold-sensitivity phenotypes, emphasizing that ub-hotspots are highly specific for Slx5 (Fig. R2C-G).

Figure R2. (A) ChIP-qPCR analysis of ubiquitin enrichments in *slx5Δ slx8Δ* strains with plasmid-borne expression of mammalian STUbL variants (Uzunova *et al*, 2007; Sun *et al*, 2014). (B) ChIP-qPCR for ubiquitin in the indicated strains. For comparison with (A), the same scale on the Y-axis was used. (C) Plasmid-based complementation of *slx5Δ* strains with Slx5-RNF4/Arkadia hybrid constructs as indicated. (D-F) ChIP-qPCR quantification for enrichments of ubiquitin (D), Euc1 (E) or hybrid STUbL constructs (using polyclonal anti-Slx5 or anti-FLAG antibodies) (F). (G) Expression levels of hybrid STUbL constructs as probed by WB using a polyclonal anti-Slx5 antibody.

Minor comments:

1) I'm not so fond of the part of the discussion about 'SUMO-SIM interaction only' mode. Data showing that a STUbL-sim mutant does not complement a certain phenotype does not rule out other substrate-recognition features involved in a particular interaction. In the manuscript, both the Slx5-SIM* and Slx5-MdD mutants appear to be fully defective for ub-hotspot formation (Figure 5). It may be that most STUbL functions involve additional interactions other than SUMO-SIM interactions, as alluded to in the final paragraph of the discussion. Simply based on protein size, I would think that Slx5/8 and Arkadia have more macromolecular interactions than RNF4. I would change the wording at the end of the discussion a bit, maybe removing the "only" and focusing on the SIM-independent interactions by STUbLs, with Euc1 being another example.

We agree and have modified the discussion accordingly, we have also replaced 'SUMO-SIM interaction only' mode by 'polySUMO-SIM interaction mode'. We think that it is entirely possible that for other Slx5/Slx8 substrates additional substrate recognition features exist. It is, however, remarkable that complementation of *slx5*Δ HU-sensitivity is possible by a variety of STUbL architectures as long as SIMs and a functional RING-domain are present (see also Fig. R2).

REFERENCES

Costanzo et al. (2010) *Science*. 327 (5964): 425-31.

Costanzo et al. (2016) *Science*. 353 (6306). pii:aaf1420

van de Pasch et al. (2013) *PLoS One*. 8 (6): e65628

Yu et al. (2008) *Science*. 322 (5898):104-10.

Zheng et al. (2010) *Mol Syst Biol*. 6: 420

Reviewer #3:

To investigate Slx5/Slx8 STUbL-mediated ubiquitylation of chromatin-associated proteins in budding yeast, the authors carried out ChIP-seq for Ub itself and Slx8. Ub signals were found widely across the genome, with signals being increased in a *cdcd48-6* mutant background, presumably as a result of increased H2B-Ub levels. In contrast, Slx8 was only localized to a few sites, overlapping with strong Ub signals at seven discrete intergenic sites on four different chromosomes. They showed that K48 Ub chains were present at these sites, and that this was dependent on functional Slx5 and Slx8. Analysis of the seven sites revealed a conserved 36 bp motif, and using a 1-hybrid screen to identify motif-binding proteins they identified YMR11, which they renamed EUC1 (Enriches Ubiquitin on Chromatin), as well as SLX5 and SMT3/SUMO. Using anti-Euc1 antibodies for ChIP-seq they showed that Euc1 was primarily located at these seven hotspots, and that mutation of the consensus motif abolished Euc1 binding. Euc1 possesses a Gcr1 domain at the C-terminus, which is known to have sequence-specific DNA-binding activity, and they showed that Euc1 could drive expression of a HIS3 reporter gene equipped with the binding motif upstream of HIS3 gene; reduced reporter activity was noted in a *euc1Δ* strain. However, expression of the genes surrounding the chromosomal hotspots was unaffected in the *euc1Δ* strain. They mapped a transactivation domain in Euc1 to an acidic region containing residues 19-28. They showed that Euc1 was sumoylated predominantly at K231, lying in a SUMO consensus motif, generating mono-sumoylated Euc1 species, and that Ub-hotspot formation did not occur in a K231R Euc1 strain. A Siz1/Siz2 double knockout strain lacked Ub hotspots, suggesting that Siz1/Siz2 are the major Euc1 SUMO E3 ligases. Euc1 was shown to be polyubiquitylated in a Slx5/Slx8-dependent manner, consistent with the ubiquitylation being carried out by Slx5/Slx8-mediated STUbL activity, and the demonstration that Euc1 was required for Slx5/Slx8 recruitment to the hotspots. Slx5/Slx8 recruitment to substrates is often dependent on interaction of its tandem SIMs with SUMO chains, but Euc1 is only mono-sumoylated and they showed that in addition to the Slx5 SIM interaction with SUMO-K231 there is also a direct interaction of Euc1 and Slx5 between a region containing a coiled-coil dimerization motif and downstream sequences in the middle of Euc1, and the M-domain lying between the SIMs and the RING domain in Slx5. Mutations in Euc1 that abolished Slx5 binding reduced ubiquitylation of Euc1 itself and also the formation of the Ub hotspots. Overall, they conclude that a multivalent interaction between Euc1 and Slx5 mediates Slx5/Slx8 recognition of Euc1 as a SUMO substrate for ubiquitylation.

This is a very nice paper showing that the Euc1 transcription factor binds to a minimal set of discrete sites at seven chromosomal locations in the *S. cerevisiae* genome. Euc1 bound at these sites is in turn sumoylated by the Siz1/Siz2 SUMO ligases, resulting in the recruitment of the Slx5/Slx8 STUbL, leading to local polyubiquitylation of Euc1 and other chromatin components. The data are convincing, but what is missing is the true functional significance of this series of events at these seven chromosomal Ub hotspots, i.e. what is the physiological function of these Ub hotspots. No strong phenotype was noted with the *euc1Δ* strain, or a significant difference in the transcription of genes adjacent to the hotspots.

Points: 1. To what extent are Euc1 itself and the hotspots conserved in other yeasts or in higher eukaryotes?

We apologize that our initial manuscript failed to comment on the aspect of evolutionary conservation, we now address this in Fig. 2I-J and Fig. S3G-H. Briefly, clear Euc1-homologs can be easily detected in several other yeast species and conservation is most striking in the DNA-binding domain and around the coiled-coil domain. In related *Saccharomyces* species we can also observe the ub-hotspot motif and we found at least one ub-HS-motif for each ub-hotspot, with particular conservation of the consensus motif. In more distantly related species it becomes difficult to prove the presence of ub-HS motifs as genome structure is diverging. We therefore cannot comment on conservation in higher eukaryotes.

2. What are the phenotypes of *euc1*Δ cells, for instance when they are subjected to stress?

We could not observe any strong growth phenotype of *euc1*Δ cells, when we tested conditions such as cold/heat stress, oxidative stress, reducing agents, growth on various different carbon sources, DNA-damage inducing agents, ER protein folding stress, hypoxia, HDAC-inhibitors, drugs targeting lipid biosynthesis and others.

However, when combining *euc1*Δ with several mutants in chromatin remodelers and regulators of gene expression such as the H2A.Z-SWR1C-pathway, *NPL3*, *STB5* and the Rpd3L complex (*DEP1*, *RXT2* and *SDS3*), this revealed strong growth phenotypes on conditions such as thermostress (cold or heat), DMSO or hydroxyurea (Fig. 7A-C, S11). Importantly, our genetic analysis places *EUC1* in a common pathway with members of the Rpd3S complex (*RCO1*, *EAF3*, Fig. 7D-E).

See also main text and reply to Reviewer 2, point (1) for more details.

3. Page 7, line 223: Here, the text and the data in Figure S3G do not seem consistent with each other. Deletion of *ubx5* conferred a mild activation of the *HIS3* reporter gene expression (although a WT cell control should be shown), which was abolished by *euc1* deletion. Here, an experiment using the same ub-HS4-*HIS3* reporter assay, comparing WT with *euc1*-KR, *siz1*Δ *siz2*Δ, *slx5*Δ, *slx8*Δ, *cdc48-3*, *euc1*Δ, as well as *euc1*Δ combined with *siz1*, *siz2*, *slx5*, *slx8*, and *cdc48* mutants would be informative. If the model is correct, *HIS3* expression may be turned on when the pathway is disrupted, and then turned back off when *euc1* is deleted. This result would be a good addition to support their proposed model regarding its physiological function.

We apologize if our description for Fig. S3G (now Fig. S4A) was unclear and clarified the text accordingly. Actually, we did not aim to show activation caused by *ubx5*Δ in this experiment, but rather made the observation that ‘background’ (i.e. without expression of an additional AD-containing construct) activation of *HIS3* was abolished in the *3x ub-HS4-HIS3 ubx5*Δ Y1H reporter strain when *EUC1* was deleted.

We addressed the reviewer’s suggestion by creating *euc1*Δ Y1H reporter strains with deletions of Euc1-ub-hotspot pathway enzymes that we complemented with *EUC1*-plasmids (Fig. R3). In the original *3x ub-HS4-HIS3* strain, deletion of *SLX5* and *SLX8* caused a defect for activation by Euc1, but not Euc1-KR (Fig. R3A). We note however that a different reporter construct, a single copy of *ub-HS5* with 100bp flanking regions, yielded different results. Here, abolishing Euc1-SUMOylation by introducing the K231R mutation or deleting *SIZ1* and *SIZ2* led to a clear activation (Fig. R3B), similar to what we had observed for the Euc1¹⁻²⁹⁵-Gal4-BD constructs (now Fig. S9B). Deletion of *SLX5* or *SLX8* had little effect and *ubx5*Δ (which we chose over *cdc48-ts* mutants for growth-based assays) had no effect in both strains.

We also tested other Euc1-constructs for activation of both reporter gene constructs and found that Euc1-KR-SBM1+2* seemed to activate best (Fig. R3C-D). As SUMO had a variable effect on transcriptional activation with different reporters we decided not to present a firm conclusion in the manuscript, particularly because we think that Euc1-mediated transcriptional activation is not critical for Euc1 function at hotspots.

Instead, we would like to highlight that complementation of *euc1Δ* defects in *sds3Δ* and *npl3Δ* fully depended on the ability of the *euc1*-alleles to form ub-hotspots, but not on the N-terminal transactivation domain (N30Δ, Fig.7G, S13A). Therefore, a major function of Euc1 in these situations seems to be formation of ub-hotspots, but not transactivation.

Figure R3. (A) Reporter gene assays using the *3x ub-HS4-HIS3* reporter in the indicated genetic backgrounds. All strains have *EUC1* deleted and were complemented with plasmids as indicated on the top. (B) As (A), but using a different reporter construct featuring *ub-HS5* with 100bp genomic context flanking on both sides. (C) The reporter strain used in (A, top row 'WT + *euc1Δ*') was transformed with plasmids expressing the indicated Euc1-constructs. (D) As in (C), but with the *ub-HS5* reporter strain.

4. Does n-terminal fusion of SUMO to K231R-Euc1 rescue the hotspot defect?

We created both N- and C-terminal fusions of SUMO with Euc1, however, none of them rescued the ub-hotspots or loss of binding of *euc1-KR* to ub-hotspots (Fig. R4A-B). Rather, several different covalent SUMO-fusions were undetectable by Western blotting (Fig. R4C). At this point, we cannot be sure whether these fusion constructs are turned over very quickly or whether cells fail to express these constructs for other reasons.

Figure R4. (A-B) ChIP-qPCR experiments of ubiquitin and Euc1 for Euc1-SUMO fusion constructs as indicated. (C) WB of total protein extracted from cells used in (A-B). Note that for N-terminal fusions we used Smt3-Q95P (analogous to the non-deconjugatable SUMO2-Q90P (Békés *et al.*, 2011)) and Smt3-AA for C-terminal fusions (C-terminal di-glycine mutated to alanines).

5. Is anything known about what de-sumoylates SUMO-Euc1 - Ulp1/2?

We have not systematically addressed deSUMOylation, but since both Ulp1 and Slx5/Slx8 were reported to associate with components of the nuclear pore complex (NPC, Ulp1 with Mlp1/Mlp2 (Zhao *et al.*, 2004), Slx5/Slx8 with the Nup84p complex (Nagai *et al.*, 2008)), we investigated whether Ulp1 mislocalization or disruption of the Nup84p complex would cause changes to ub-hotspots or Euc1-SUMOylation (Fig. R5). Ulp1-mislocalization (Δ N-Ulp1, *mlp1/2Δ* + GFP-Ulp1) had no effect on ub-hotspots or Euc1-SUMOylation, but curiously, a Gfp1-Nsp1-Ulp1 construct that reportedly targets mislocalized Ulp1 back to the NPC (Texari *et al.*, 2013) caused defects in ub-hotspots, Euc1 binding and Euc1-SUMOylation (Fig. R5A-C).

Disruption of the Nup84p complex (*nup84Δ*, *nup133Δ*) also led to strong defects in ubiquitylation levels at ub-hotspots (Fig. R5D-E), but interestingly, Euc1 enrichments were barely affected and Euc1 SUMOylation even increased (Fig. R5F).

These data support our previous notion that Euc1-SUMOylation is required for ubiquitylation, but also stabilizing Euc1-binding to ub-hotspots. We speculate that Nup84p disruption might lead to Slx5/Slx8 impairment causing the observed ubiquitylation defects. Alternatively, it is possible that ub-hotspots might require association with NPCs for ubiquitylation. We note, however, that we could not detect any crosslinking of Nup84 or Nup133 to ub-hotspots in ChIP experiments.

Figure R5. (A-B) ChIP-qPCR analysis of ubiquitin (A) and Euc1 (B) enrichment at selected sites in Ulp1-mislocalization mutants as published in (Texari *et al*, 2013). Note that in ΔN -*Ulp1-GFP* and $\Delta ulp1 \Delta mlp1/2 + GFP-Ulp1$ cells Ulp1 localization to nuclear pores is lost, while fusion to Nsp1 (*GFP-NSP1-Ulp1*) redirects Ulp1 to nuclear pores (Texari *et al*, 2013). (C) WB analysis of immunoprecipitated Euc1 from strains used in (A-B). (D-F) ChIP-qPCR for ubiquitin or Euc1 in mutants of the NPC complex Nup84p (*nup84Δ*, *nup133Δ*). (G) Immunoprecipitated Euc1 from Nup84p-mutant strains was analyzed by WB.

6. Could Euc1 drive transcription of a non-coding RNA?

Indeed, we considered this possibility, especially because some ub-hotspots overlap with annotated non-coding (nc) RNAs, including some Xrn1-sensitive transcripts (XUTs (van Dijk *et al*, 2011)), cryptic unstable transcripts (CUTs (Wyers *et al*, 2005; Xu *et al*, 2009)), stable unannotated transcripts (SUTs, (Xu *et al*, 2009)) and Nrd1-untersminated transcripts (NUTs (Schulz *et al*, 2013)). In an RNA sequencing experiment, we specifically investigated XUTs and CUTs in WT and RNA surveillance mutant cells (*xrn1Δ rrp6Δ*). We compared ncRNA abundance in both backgrounds with WT and DNA-binding deficient Euc1 (*euc1-DBD**), however, only minor changes at only some ub-hotspots were detected.

References

- Békés M, Prudden J, Srikumar T, Raught B, Boddy MN & Salvesen GS (2011) The Dynamics and Mechanism of SUMO Chain Deconjugation by SUMO-specific Proteases. *J. Biol. Chem.* **286**: 10238–10247
- Nagai S, Dubrana K, Tsai-Pflugfelder M, Davidson MB, Roberts TM, Brown GW, Varela E, Hediger F, Gasser SM & Krogan NJ (2008) Functional targeting of DNA damage to a nuclear pore-associated SUMO-dependent ubiquitin ligase. *Science* **322**: 597–602
- Ohkuni K, Takahashi Y, Fulp A, Lawrimore J, Au W-C, Pasupala N, Levy-Myers R, Warren J, Strunnikov A, Baker RE, Kerscher O, Bloom K & Basrai MA (2016) SUMO-Targeted Ubiquitin Ligase (STUbL) Slx5 regulates proteolysis of centromeric histone H3 variant Cse4 and prevents its mislocalization to euchromatin. *Mol. Biol. Cell* **27**: 1500–1510
- Prudden J, Pebernard S, Raffa G, Slavin DA, Perry JJP, Tainer JA, McGowan CH & Boddy MN (2007) SUMO-targeted ubiquitin ligases in genome stability. *EMBO J.* **26**: 4089–4101
- Schulz D, Schwalb B, Kiesel A, Baejen C, Torkler P, Gagneur J, Soeding J & Cramer P (2013) Transcriptome surveillance by selective termination of noncoding RNA synthesis. *Cell* **155**: 1075–1087
- Schweiggert J, Stevermann L, Panigada D, Kammerer D & Liakopoulos D (2016) Regulation of a Spindle Positioning Factor at Kinetochores by SUMO-Targeted Ubiquitin Ligases. *Dev. Cell* **36**: 415–427
- Sun H, Levenson JD & Hunter T (2007) Conserved function of RNF4 family proteins in eukaryotes: targeting a ubiquitin ligase to SUMOylated proteins. *EMBO J.* **26**: 4102–4112
- Sun H, Liu Y & Hunter T (2014) Multiple Arkadia/RNF111 Structures Coordinate Its Polycomb Body Association and Transcriptional Control. *Mol. Cell. Biol.* **34**: 2981–2995
- Texari L, Dieppois G, Vinciguerra P, Contreras MP, Groner A, Letourneau A & Stutz F (2013) The nuclear pore regulates GAL1 gene transcription by controlling the localization of the SUMO protease Ulp1. *Molecular Cell* **51**: 807–818
- Uzunova K, Götttsche K, Miteva M, Weisshaar SR, Glanemann C, Schnellhardt M, Niessen M, Scheel H, Hofmann K, Johnson ES, Praefcke GJK & Dohmen RJ (2007) Ubiquitin-dependent proteolytic control of SUMO conjugates. *J. Biol. Chem.* **282**: 34167–34175
- van Dijk EL, Chen CL, d'Aubenton-Carafa Y, Gourvenec S, Kwapisz M, Roche V, Bertrand C, Silvain M, Legoix-Né P, Loeillet S, Nicolas A, Thermes C & Morillon A (2011) XUTs are a class of Xrn1-sensitive antisense regulatory non-coding RNA in yeast. *Nature* **475**: 114–117
- Wyers F, Rougemaille M, Badis G, Rousselle J-C, Dufour M-E, Boulay J, Régnauld B, Devaux F, Namane A, Séraphin B, Libri D & Jacquier A (2005) Cryptic pol II transcripts are degraded by a nuclear quality control pathway involving a new poly(A) polymerase. *Cell* **121**: 725–737

Xu Z, Wei W, Gagneur J, Perocchi F, Clauder-Munster S, Camblong J, Guffanti E, Stutz F, Huber W & Steinmetz LM (2009) Bidirectional promoters generate pervasive transcription in yeast. *Nature* **457**: 1033–1037

Zhao X, Wu C-Y & Blobel G (2004) Mlp-dependent anchorage and stabilization of a desumoylating enzyme is required to prevent clonal lethality. *J. Cell Biol.* **167**: 605–611

Thank you again for submitting a new version of your manuscript on STUbL recruitment and Euc1-related ubiquitin hotspots on chromatin for our consideration. I sent it back to the original referees 2 and 3, and I am pleased to inform you that both of them consider the study significantly extended and, despite some remaining uncertainty about the roles of Euc1, now suitable for publication in The EMBO Journal. Following some minor editorial revisions as listed below, we shall therefore be happy to accept the paper for publication!

In the course of this final minor revision, please incorporate the remaining comment from referee 2 and address the following editorial points.

REFeree REPORTS.

Referee #2:

This revised paper has exactly what was missing in the original, otherwise excellent version of the manuscript, namely, evidence for a functional role of the Slx5/Slx8-regulated ubiquitin hotspots in the yeast genome (new Figure 7). Mechanistically, it is still not clear what the Euc1 protein is doing exactly, but overall, this is very nice study, and I feel it is suitable for publication in the EMBO Journal in its present form.

One tiny point from the Intro:

Line 57: "However, in case of the prototypical STUbL, budding yeast Slx5/Slx8, no in vivo evidence has been found for SUMO-SIM-independent substrate recognition." This is a little misleading because as pointed out later in the paper, SUMO independence had been found for at least one yeast substrate (Xie et al., 2010).

Referee #3:

The authors have strengthened the paper by including some functional evidence in the new Figures 6 and 7, where they showed that in combination with LOF mutations in *htz1*, which encodes histone H2A.Z, or in subunits of the Rfp3 HDAC, (*dep1Δ*, *rxt2Δ*, *sds3Δ*) a *euc1* LOF mutation causes a synthetic sick phenotype, particularly when the cells are subject to heat or DMSO stresses, and they provide evidence that the Euc1 and Rfp3 are epistatic. Nevertheless, in the end it remains unclear what Euc1 does, since the stress resistance function does not require the putative N-terminal transactivation domain, whereas it does require Euc1 DNA binding activity, i.e. why does it have to be bound to these genomic loci (what would happen if one of these hotspots was moved to another genomic location?). They also showed that overexpression of Euc1 is toxic, in a manner exacerbated by *slx5Δ* or *slx8Δ* loss. In summary, the conclusion that Slx5/Slx8 STUbL function is required to regulate Euc1 activity at these Ub hotspots under stressful conditions is reasonable, but in the end it is not exactly clear what this activity is, even though the function is apparently conserved in other yeasts. Nevertheless, the existence of these genomic hotspots for ubiquitylation is intriguing and the revised version contains a lot of new data that provide some functional clues. Scientifically, this is solid, I would now be in favor of accepting this version for EMBO Journal

We would like to thank all reviewers for their time and constructive criticism towards our paper. We think that the review process greatly helped us to improve our paper.

Referee #2:

This revised paper has exactly what was missing in the original, otherwise excellent version of the manuscript, namely, evidence for a functional role of the Slx5/Slx8-regulated ubiquitin hotspots in the yeast genome (new Figure 7). Mechanistically, it is still not clear what the Euc1 protein is doing exactly, but overall, this is very nice study, and I feel it is suitable for publication in the EMBO Journal in its present form.

One tiny point from the Intro:

Line 57: "However, in case of the prototypical STUbL, budding yeast Slx5/Slx8, no in vivo evidence has been found for SUMO-SIM-independent substrate recognition." This is a little misleading because as pointed out later in the paper, SUMO independence had been found for at least one yeast substrate (Xie et al., 2010).

We thank the reviewer for his/her efforts and have changed the corresponding passage in the paper, to highlight our main point, namely that before our work no SUMO-independent interaction surfaces have been characterized in the STUbL Slx5/8.

Referee #3:

The authors have strengthened the paper by including some functional evidence in the new Figures 6 and 7, where they showed that in combination with LOF mutations in *htz1*, which encodes histone H2A.Z, or in subunits of the Rfp3 HDAC, (*dep1Δ*, *rxl2Δ*, *sds3Δ*) a *euc1* LOF mutation causes a synthetic sick phenotype, particularly when the cells are subject to heat or DMSO stresses, and they provide evidence that the Euc1 and Rpf3 are epistatic. Nevertheless, in the end it remains unclear what Euc1 does, since the stress resistance function does not require the putative N-terminal transactivation domain, whereas it does require Euc1 DNA binding activity, i.e. why does it have to be bound to these genomic loci (what would happen if one of these hotspots was moved to another genomic location?). They also showed that overexpression of Euc1 is toxic, in a manner exacerbated by *slx5Δ* or *slx8Δ* loss. In summary, the conclusion that Slx5/Slx8 STUbL function is required to regulate Euc1 activity at these Ub hotspots under stressful conditions is reasonable, but in the end it is not exactly clear what this activity is, even though the function is apparently conserved in other yeasts. Nevertheless, the existence of these genomic hotspots for ubiquitylation is intriguing and the revised version contains a lot of new data that provide some functional clues. Scientifically, this is solid, I would now be in favor of accepting this version for EMBO Journal

We thank the reviewer for the constructive criticism and that he/she supports publication of our paper.

Accepted

3rd April 2019

Thank you for submitting your final revised manuscript for our consideration. I am pleased to inform you that we have now accepted it for publication in The EMBO Journal.

Corresponding Author Name: Pfander, Boris

Journal Submitted to: The EMBO Journal

Manuscript Number: EMBOJ-2018-100368R